# Recall Residualization: Decontaminating Foundation-Model Evaluation on Public Time-Series Benchmarks

**Anany Kotawala** [1]

## Abstract

Foundation-model evaluations on public time-series benchmarks risk conflating forecasting skill with parametric recall: factor returns, macroeconomic indicators, and climate indices are widely mirrored in pretraining data, so a model's apparent "skill" on such a benchmark can be retrieval of labels rather than predictive ability. Standard evaluation cannot tell the two apart. We propose *Recall Residualization*: regress any LLM-derived signal on the model's own direct value-recall of the benchmark, and report the residual signal–truth correlation as decontaminated skill. A worst-case orthogonal decomposition gives an upper bound; OLS gives an exact point estimate. In a 20-cell study across four benchmarks and five frontier LLMs, apparent skill up to $|\rho|=0.93$ collapses to $|\rho| \leq 0.17$ after residualization (LeakShare $\geq 0.97$ in $19/20$ cells). Dedicated time-series foundation models forecast the same months at AR(1)-level performance or below under their native numeric-history interface: the contamination channel is interface-specific to date- and label-conditioned text queries. Code here: https://github.com/ananykotawala/recall-residualization.

## 1. Introduction

Public time-series benchmarks create a distinctive contamination risk for foundation-model evaluation: their labels are ordered numeric records that are widely mirrored online. Foundation-model evaluations increasingly rely on public benchmark datasets whose values are not text passages but ordered numeric records: prices, macro releases, climate indices, factor libraries. These series underpin both classical pipelines and the growing class of dedicated structured-

data foundation models (Ansari et al., 2024; Garza and Mergenthaler-Canseco, 2023), and they are widely mirrored, documented, and reused. If a foundation model can recover historical labels from a date and a public series name, any downstream signal it produces can inherit those labels: an apparent correlation between the signal and the truth may be skill, recall, or both, and standard evaluation cannot tell them apart.

We propose **Recall Residualization** as a decontamination tool. Given a model $M$, public series $r$, and an $M$-derived evaluation signal $S$, we elicit $M$'s direct recall $\hat{r}$ of the same series and dates, regress $S$ on $\hat{r}$, and report the residual signal–truth correlation $\rho(u, r)$ as the decontaminated score. The procedure converts "is this skill or recall?" from an unanswerable concern into a measurement.

Recent contamination and LLM-finance work raises related concerns about memorized economic knowledge, look-ahead bias, and benchmark leakage (Magar and Reichart, 2022; Carlini et al., 2023; Lopez-Lira et al., 2025; Li et al., 2025; Benhenda, 2026); closest to our setting, Lopez-Lira et al. (2025) study memorization in economic forecasting. Our contribution is methodological: rather than auditing whether contamination occurred, we provide a procedure that decontaminates downstream evaluations directly. Recall Residualization applies to any model class that accepts a date- and label-conditioned query and emits a numeric estimate; it is empty for models without that interface (verified on TS-FMs in §3.3).

**Contributions.**

- We propose *Recall Residualization*, a decontamination procedure that regresses an LLM-derived signal on the model's own value-recall of the benchmark and reports the residual signal–truth correlation as a decontaminated score. The procedure has a worst-case orthogonal-decomposition ceiling and an exact OLS point estimate when recall is co-locatable, with bootstrap CIs and a single-number leak-share metric.

- We demonstrate the procedure across four public time-series benchmarks (Fama–French Mkt-RF, FRED UN-RATE, FRED CPI YoY, NASA GISTEMP) and 20 model×signal cells on five frontier LLMs spanning

[1]Princeton University, Princeton, NJ, USA. Correspondence to: Anany Kotawala <akotawala@princeton.edu>.

*Accepted at the 2nd ICML 2026 Workshop on Foundation Models for Structured Data (FMSD)*, Seoul, South Korea.

four vendors (Anthropic, OpenAI, Meta, DeepSeek): residualization collapses apparent skill from $|\rho| \in [0.26, 0.93]$ to $|\rho| \leq 0.17$ with LeakShare $\geq 0.97$ in 19/20 cells.

- We scope the contamination channel to interface: five TS-FM variants spanning three families (Chronos (Ansari et al., 2024), TimesFM-2.0 (Das et al., 2024), Moirai-1.0-R-large (Woo et al., 2024)) under their native numeric-history forecasting interface forecast the same months at AR(1)-level performance or below, locating the contamination at the query-format boundary rather than the foundation-model substrate.

We close in §4 with the procedure's scope and limitations, broader implications for benchmark-driven foundation-model evaluation, and natural extensions.

## 2. Method: Recall Residualization

Let $r_t$ denote the public value of a series $r$ at month $t$, and let $S_t$ be a date-conditioned signal produced by a foundation model $M$ on the same months (e.g., a sentiment score, a forecast, a class label). Standard evaluation reports $\rho(S, r)$ as the model's apparent skill on the benchmark; this conflates genuine skill with parametric recall of the labels.

**Direct recall.** We elicit the model's recall of the benchmark via an ordinary text query $q(r, t)$ that names the series and date and asks for the value (templates in App. Q). The parsed response $\hat{r}_t$ is the model's accessible recall: an API-boundary estimand of what $M$ can produce when asked, not a claim about training-set membership or internal storage.

**Residualization.** On co-located months where both $S_t$ and $\hat{r}_t$ are available, fit OLS $S_t = \alpha + \beta\hat{r}_t + u_t$—a standard partial-regression construction (Lovell, 1963)—and define the decontaminated score

$$\rho_{\text{decon}} := \rho(u, r), \quad \text{LeakShare} := 1 - \frac{\rho_{\text{decon}}^2}{\rho_{\text{app}}^2}, \quad (1)$$

where $\rho_{\text{app}} := \rho(S, r)$ is the apparent score. LeakShare is the fraction of squared apparent correlation that the recall channel explains. Bootstrap CIs are computed by resampling co-located months.

**Worst-case ceiling.** When $S$ and $\hat{r}$ are not co-located (e.g., the auditor only has $\rho(S, r)$ from a published table), a worst-case orthogonal decomposition of $S$ on $\hat{r}$ bounds the leak component:

$$\rho_{\text{leak,max}} = \min\left(1, \frac{|\rho_{\text{recall}}|}{|\rho_{\text{app}}|}\right) \cdot \rho_{\text{app}}, \quad (2)$$

where $\rho_{\text{recall}} := \rho(\hat{r}, r)$ is recall fidelity. Eq. 2 is conservative: it cannot certify any skill when $|\rho_{\text{recall}}| \geq |\rho_{\text{app}}|$ (App. P).

**Method-class scope.** Recall Residualization requires that $M$ accept a date- and label-conditioned query and emit a numeric estimate. Text-LLMs accept this interface natively. Dedicated time-series foundation models (TS-FMs) typically take a numeric history without a date or label and return a forecast, so they have no comparable recall channel to subtract; we verify this on five TS-FM variants spanning three families (Chronos, TimesFM-2.0, Moirai) in §3.3. Classical forecasters (ARIMA, exponential smoothing) have no learned parametric memory of public labels and need no decontamination.

**Algorithm.** Algorithm 1 states the procedure; the reference implementation (App. S) provides the recall and signal probes, the OLS step, bootstrap CIs, and a method-class compatibility classifier. All experiments use ordinary no-tool API calls with temperature 0 where supported, and Claude Opus 4.7 is called without the unsupported temperature parameter.

---

**Algorithm 1** Recall Residualization.

---

1: **input:** model $M$, public series $r$ over months $\mathcal{T}$, signal prompt $\sigma$, value prompt $\nu$
2: $\mathcal{D} \leftarrow \{\}$
3: **for** $t \in \mathcal{T}$ **do**
4:     $S_t \leftarrow \text{parse}(M(\sigma(t)))$          // signal
5:     $\hat{r}_t \leftarrow \text{parse}(M(\nu(r, t)))$     // recall
6:     **if** $S_t \neq \bot$ **and** $\hat{r}_t \neq \bot$ **then**
7:         $\mathcal{D} \leftarrow \mathcal{D} \cup \{(t, S_t, \hat{r}_t, r_t)\}$
8:     **end if**
9: **end for**
10: Fit OLS on $\mathcal{D}$: $S_t = \alpha + \beta\hat{r}_t + u_t$
11: $\rho_{\text{decon}} \leftarrow \rho(u, r)$
12: $\rho_{\text{app}} \leftarrow \rho(S, r)$
13: **report** parsed-coverage $|\mathcal{D}|/|\mathcal{T}|$
14: **return** $\rho_{\text{decon}}, 1 - \rho_{\text{decon}}^2/\rho_{\text{app}}^2$    // LeakShare

---

## 3. Results

### 3.1. Recall is widespread on public time-series benchmarks

Recall Residualization presupposes that the model has measurable recall of the benchmark to subtract. We verify this on four public time-series benchmarks spanning finance (Fama–French Mkt-RF (French, 2026)), labor (FRED UN-RATE), prices (FRED CPI YoY percent change), and climate (NASA GISTEMP land–ocean global monthly temperature anomaly). For each, we issue an ordinary value-recall query naming the series and date and parse a signed numeric

value; we report Pearson $r$ against published truth on parsed responses.

*Table 1.* **Recall fidelity on four public time-series benchmarks × five frontier LLMs from four vendors.** $\rho_{\text{recall}}$ is the Pearson correlation between the model's parsed value-query response and the published truth on the matched-month subset (§3.2). Frontier LLMs recall published values at near-truth correlation across finance, macro, and climate domains, supplying the input that Recall Residualization requires; recall is weaker for Llama-4-Maverick and DeepSeek-V3.2 on Mkt-RF. *Vendor versions:* the residualization study uses GPT-5.5 (current frontier OpenAI tier) and Llama-4-Maverick; appendix recall sweeps additionally cover the earlier GPT-5.4 family and Llama-3.3-70B / 3.1-8B on Mkt-RF only (App. B).

| Series | Src | $n$ | Sonnet | Opus | GPT-5.5 | Llama-4 | DeepSeek |
|---|---|---|---|---|---|---|---|
| Mkt-RF | FF | 35–40 | +0.95 | +0.98 | +0.99 | +0.57 | +0.47 |
| UNRATE | FRED | 30 | +1.00 | +1.00 | +1.00 | +1.00 | +1.00 |
| CPI YoY | FRED | 29–30 | +1.00 | +1.00 | +1.00 | +0.98 | +1.00 |
| GISTEMP | NASA | 30 | +0.92 | +0.93 | +1.00 | +0.84 | +0.93 |

Within Fama–French specifically, recall is selective: the aggregate-market factor (Mkt-RF) is recovered at $r\approx0.98$ while the five style factors (SMB, HML, RMW, CMA, Mom) sit far below, and a factor-shuffle null is $\sim19\times$ lower than observed Mkt-RF recall on Sonnet (App. B). Recall survives prompt rewording (App. L), is stable in the answered pre-cutoff region (App. E), and replicates across providers and capability tiers (App. F). A label-invariance probe shows the recalled object is an aggregate-market public series rather than a literal Fama–French label artifact: removing the FF name preserves recall on S&P 500 and "U.S. market excess" (App. G). As a negative control, Anthropic models that recall Mkt-RF refuse syntactically identical fictional-factor prompts in $180/180$ cases, ruling out generic numeric fluency (App. M).

The recall channel is interface-specific: under text-conditioned date/label queries, frontier LLMs recover Mkt-RF at $|\rho| \geq 0.95$ on the top tier (Tab. 1); under numeric-history forecasting interfaces, the five TS-FM variants tested in §3.3 (full table in App. U) forecast the same months at AR(1)-level performance or below (best: Moirai-1.0-R-large at $\rho=+0.243$ vs. AR(1) at $\rho=+0.238$).

### 3.2. Decontamination via Recall Residualization

The headline application: take a date-conditioned signal $S$ that the model produces in response to an indirect query (a sentiment-style narrative prompt, not a value query), measure its apparent correlation $\rho_{\text{app}}$ with the public truth, then apply Eq. (1) to obtain the decontaminated $\rho_{\text{decon}}$. The signals are: *Mkt-RF*, an investor sentiment in $[-1, +1]$ for U.S. equities; *UNRATE*, a labor-market condition score in $[-1, +1]$; *CPI YoY*, an inflation-pressure score in $[-1, +1]$; *GISTEMP*, a global temperature condition score in $[-1, +1]$. None of these prompts asks for a value (full templates in

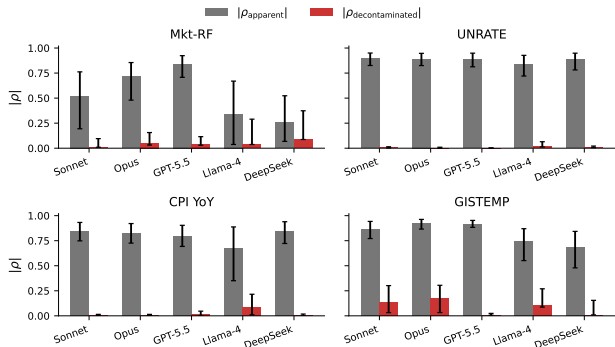

*Figure 1.* Apparent vs. decontaminated $|\rho|$ across all 20 model×signal cells. Recall Residualization collapses apparent skill in every cell. Error bars are 95% bootstrap CIs. The small residual on GISTEMP reflects multi-timescale recall that the linear adapter underfits (App. T).

App. Q); per-cell results with bootstrap CIs are in Tab. 2, summarised graphically in Fig. 1.

In every cell, residualization eliminates almost all of the apparent signal–truth correlation. On the three economic benchmarks the residual is small in 13 of 15 cells (Leak-Share $\geq 0.99$); the exceptions, Mkt-RF / DeepSeek-V3.2 and CPI YoY / Llama-4-Maverick, both have weaker recall fidelity (Tab. 1). On GISTEMP small positive residuals survive (largest $+0.17$), reflecting a multi-timescale recall channel that the linear adapter only partially captures (App. T). The interpretation is sharp: the apparent skill of an LLM-derived signal on a public time-series benchmark, when the model can recall that benchmark, is dominated by recall—not by the sentiment, narrative, or domain knowledge the prompt seems to elicit.

**Worst-case ceiling.** When co-located residualization is not available, Eq. (2) bounds the leak component using only $\rho_{\text{recall}}$ and the published $\rho_{\text{app}}$. Plugging $|\rho_{\text{app}}|\sim0.07$ from a published GPT-4 sentiment strategy (Lopez-Lira and Tang, 2023) into Eq. (2) with our observed $\rho_{\text{recall}}\geq0.27$ (every Anthropic and OpenAI tier above nano on Mkt-RF) saturates the bound at the reported alpha (App. P). This is the conservative ceiling, not the realized leak: a co-located residualization on the actual headlines and per-day GPT outputs would likely yield a substantially smaller estimate. We present the co-located residualization on our own probes as the primary contribution.

### 3.3. Interface-specific scope

Recall Residualization requires the model to accept a date- and label-conditioned query and emit a numeric estimate. We test whether dedicated time-series foundation models exhibit a recall channel under their native interface by giving each model the prior 60 months of Mkt-RF as numeric history (no series label) and asking for a one-step-ahead

*Table 2.* **Recall Residualization across four benchmarks $\times$ five frontier LLMs (Anthropic, OpenAI, Meta, DeepSeek), 20 model$\times$signal cells.** Apparent signal–truth correlation $\rho_{\text{app}}$ collapses after residualizing on the model's own recall of the same series. LeakShare is the fraction of squared apparent correlation explained by recall (Eq. 1). Brackets are 95% bootstrap CIs ($N=2000$); LeakShare CIs capped at $[0,1]$ since values $> 1$ are not meaningful (occurs in low-$\rho_{\text{app}}$ cells). $S$ prompts: Mkt-RF=investor sentiment; UNRATE=labor-market condition; CPI YoY=inflation-pressure score; GISTEMP=global temperature condition (none asks for a value; templates in App. Q).

| Series | Model | $n$ | $\rho_{\text{app}}$ [95% CI] | $\rho_{\text{decon}}$ [95% CI] | LeakShare [95% CI] |
|---|---|---|---|---|---|
| Mkt-RF | Sonnet 4.6 | 40 | $+0.52\,[+0.19,+0.76]$ | $+0.01\,[-0.07,+0.10]$ | $1.00\,[0.93,1.00]$ |
| Mkt-RF | Opus 4.7 | 40 | $+0.72\,[+0.48,+0.86]$ | $+0.05\,[-0.03,+0.16]$ | $1.00\,[0.95,1.00]$ |
| Mkt-RF | GPT-5.5 | 40 | $+0.84\,[+0.71,+0.92]$ | $+0.04\,[-0.03,+0.12]$ | $1.00\,[0.98,1.00]$ |
| Mkt-RF | Llama-4-Maverick | 35 | $+0.34\,[-0.04,+0.67]$ | $+0.04\,[-0.22,+0.29]$ | $0.99\,[0.84,1.00]$ |
| Mkt-RF | DeepSeek-V3.2 | 40 | $+0.26\,[-0.07,+0.52]$ | $+0.09\,[-0.24,+0.37]$ | $0.89\,[0.65,1.00]$ |
| UNRATE | Sonnet 4.6 | 30 | $-0.89\,[-0.95,-0.83]$ | $+0.005\,[-0.00,+0.01]$ | $1.00\,[1.00,1.00]$ |
| UNRATE | Opus 4.7 | 30 | $-0.89\,[-0.95,-0.82]$ | $-0.004\,[-0.01,+0.00]$ | $1.00\,[1.00,1.00]$ |
| UNRATE | GPT-5.5 | 30 | $-0.89\,[-0.95,-0.81]$ | $-0.001\,[-0.00,+0.00]$ | $1.00\,[1.00,1.00]$ |
| UNRATE | Llama-4-Maverick | 30 | $-0.83\,[-0.93,-0.72]$ | $+0.02\,[-0.01,+0.06]$ | $1.00\,[0.99,1.00]$ |
| UNRATE | DeepSeek-V3.2 | 30 | $-0.88\,[-0.95,-0.78]$ | $+0.005\,[-0.01,+0.02]$ | $1.00\,[1.00,1.00]$ |
| CPI YoY | Sonnet 4.6 | 30 | $+0.85\,[+0.75,+0.93]$ | $+0.004\,[-0.01,+0.01]$ | $1.00\,[1.00,1.00]$ |
| CPI YoY | Opus 4.7 | 30 | $+0.83\,[+0.73,+0.92]$ | $-0.002\,[-0.01,+0.00]$ | $1.00\,[1.00,1.00]$ |
| CPI YoY | GPT-5.5 | 30 | $+0.80\,[+0.69,+0.90]$ | $-0.02\,[-0.05,+0.00]$ | $1.00\,[1.00,1.00]$ |
| CPI YoY | Llama-4-Maverick | 29 | $+0.67\,[+0.35,+0.89]$ | $+0.08\,[-0.01,+0.22]$ | $0.99\,[0.84,1.00]$ |
| CPI YoY | DeepSeek-V3.2 | 30 | $+0.84\,[+0.72,+0.94]$ | $+0.001\,[-0.02,+0.02]$ | $1.00\,[1.00,1.00]$ |
| GISTEMP | Sonnet 4.6 | 30 | $+0.86\,[+0.77,+0.94]$ | $+0.13\,[-0.03,+0.30]$ | $0.98\,[0.88,1.00]$ |
| GISTEMP | Opus 4.7 | 30 | $+0.93\,[+0.87,+0.96]$ | $+0.17\,[+0.03,+0.30]$ | $0.97\,[0.89,1.00]$ |
| GISTEMP | GPT-5.5 | 30 | $+0.92\,[+0.88,+0.95]$ | $-0.007\,[-0.02,+0.02]$ | $1.00\,[1.00,1.00]$ |
| GISTEMP | Llama-4-Maverick | 30 | $+0.74\,[+0.55,+0.87]$ | $+0.10\,[-0.09,+0.27]$ | $0.98\,[0.87,1.00]$ |
| GISTEMP | DeepSeek-V3.2 | 30 | $+0.69\,[+0.48,+0.84]$ | $-0.009\,[-0.15,+0.13]$ | $1.00\,[0.91,1.00]$ |

*Table 3.* Interface-specific scope on the same 40 Mkt-RF months. Three TS-FM families (Chronos (Ansari et al., 2024), TimesFM (Das et al., 2024), Moirai (Woo et al., 2024)) forecast at or near AR(1)-level performance under their native numeric-history interface (Moirai's $\rho=+0.243$ vs. AR(1)'s $+0.238$ is within bootstrap noise on $n=40$); only the date- and label-conditioned text query recovers the published labels.

| Method | $\rho$ | MAE |
|---|---|---|
| Chronos-T5-base (200M) | $+0.134$ | 2.92 |
| TimesFM-2.0 (500M) | $+0.054$ | 3.07 |
| Moirai-1.0-R-large (311M) | $+0.243$ | 2.83 |
| AR(1) baseline | $+0.238$ | 2.88 |
| Opus 4.7 (text query) | $+0.99$ | 0.29 |

forecast on the same 40 Opus-probed months. Tab. 3 contrasts one representative variant from each of three TS-FM families with an AR(1) baseline and Opus's date- and label-conditioned text-query recall on the same months; the full TS-FM panel is in App. U.

Monthly Mkt-RF is near white noise, so AR(1)-level is near the achievable ceiling under a pure numeric-history interface; the contrast with text-query recall at $\rho\approx0.99$ reflects label retrieval, not TS-FM weakness. With no recall channel to subtract, residualization is a no-op for TS-FMs. Failure modes and the GISTEMP trend-vs-detrended decomposition are in App. T.

## 4. Discussion

Public benchmarks anchor foundation-model evaluation on structured data. For text-LLMs on widely mirrored time-series labels, apparent skill is dominated by parametric recall: residualizing on the model's own value-recall collapses apparent signal–truth correlation by an order of magnitude in 19 of 20 cells, across four model families and four benchmark domains. The procedure is empty for dedicated TS-FMs under numeric-history interfaces, locating the contamination at the interface boundary rather than the foundation-model substrate.

**Limitations.** Recall Residualization removes the linear component of $S$ spanned by the model's value-query recall; nonlinear or compositional contamination is captured only by the worst-case ceiling. The procedure does not certify zero contamination and does not rescue an entirely recall-driven signal. On GISTEMP, decomposing the signal into trend plus month-to-month residual shows that the non-zero $\rho_{\text{decon}}$ comes from both components, indicating a multi-timescale recall channel that the linear adapter underfits (App. T). Cost, reliability, and broader scope are in App. S and App. T.

**Implications.** We propose that tabular- or time-series foundation-model benchmarks with text-queryable labels report (apparent score, LeakShare) as a pair, with LeakShare $> 0.9$ flagged as recall-saturated. Contamination can dominate reported gains on public structured-data benchmarks; Recall Residualization converts that risk into a number. The exposure is structural, not LLM-specific: any FM evaluation whose labels are public, ordered, and text-addressable inherits the same leak channel once a wrapper LLM enters the pipeline.

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

# Appendix: Supplementary Material

## Appendix roadmap

| Appendix | Purpose | Main claim supported |
|---|---|---|
| Apps. A, B, C | Full factor/model grid, calibration, and parse rates | Aggregate-market recall is selective within the Fama–French family. |
| Apps. D, E | Cutoff and refusal controls | Answered pre-cutoff months show stable recall, while post-cutoff behavior is mainly refusal/parsability. |
| Apps. F, H, I, J | Cross-series, multi-seed, and macro replications | Numeric-series memorization is not limited to one prompt label or one finance dataset. |
| App. G | Non-LLM proxy baselines and shifted-date controls | Calendar/event priors are insufficient, while prompted-date tracking supports a date-indexed readout. |
| Apps. K, K.1, L, M | Rank/value, rewording, and fabricated-series probes | The memorization channel has distinctive behavior beyond generic numeric fluency. |
| Apps. N, O, P | Transmission probes, placebo, and leak-ceiling derivation | Recalled public labels can contaminate downstream date-conditioned signals. |
| Apps. Q, R, S | Prompt templates, sampling list, artifacts, and provenance | The protocol uses ordinary model queries and is reproducible. |
| Apps. T, V | Scope conditions and readout-entropy signature | Robustness and interpretation of the memorization channel. |

## A. Calibration grid, all 12 cells

Figure 2 is the single most informative visualization for the within-library contrast: it shows Sonnet×Mkt-RF's 45° alignment (top-left, $r=0.98$) against eleven lower-alignment cells. Points are colored by cutoff bucket; within Sonnet×Mkt-RF, pre-cutoff, near-cutoff, and post-cutoff months all land on the diagonal, supporting the claim that answered pre-cutoff months retain stable recall fidelity.

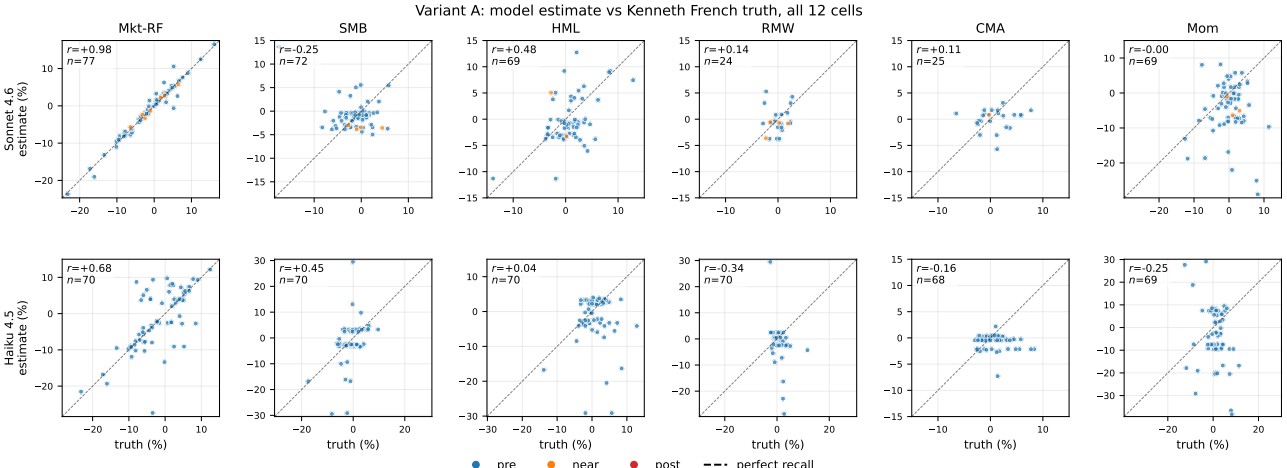

*Figure 2.* Variant A parsed estimate vs Kenneth French truth for every (model, factor) cell. Dashed line: perfect recall (45°). Annotations: Pearson $r$ and parsed-estimate count $n$ per cell. Cutoff buckets are defined in Sec. D.

## B. Per-factor headline results (full table)

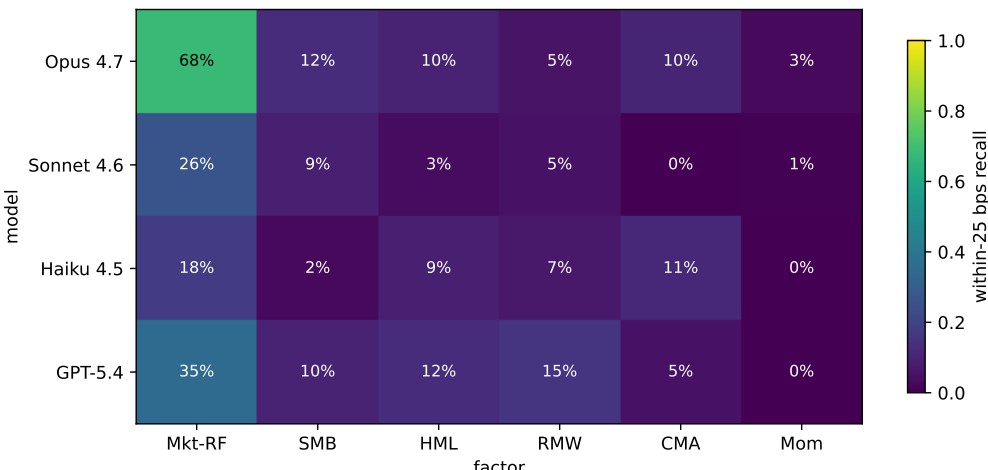

*Figure 3.* Within-25 bps recall rate per (model, factor), computed from each model's main Variant-A sweep (single-seed-42, parsed-only denominator). Mkt-RF is the only column that recovers monthly values at rates meaningfully above chance, for every model. Haiku's Mkt-RF cell (18% here) is single-seed; the honest 3-seed pooled value is 12% (see App. H / Tab. 11). Other factors stay at $\leq 15\%$ for every cell.

Table 4 reports the full 9-model $\times$ 6-factor breakdown underlying the within-family recall summary. *Provenance for Tab. 4*: Sonnet/Haiku Mkt-RF $n$ comes from the 2,784-query main sweep; Opus/GPT-5.4 from the 40-month baseline probes; the best-non-Mkt-RF row reports the factor with maximum $|r|$ per model (remaining factors are at chance, included in this full grid). The Mkt-RF column dominates everywhere; the next-most-prominent factor (SMB) shows scattered partial recall across capability tiers (Opus $r=+0.44$, Haiku $r=+0.45$, DeepSeek-V3.2 $r=+0.46$, GPT-5.4-mini $r=+0.40$) without a strict capability-tier monotone, while HML partial recall is concentrated on Opus ($r=+0.58$). RMW, CMA, and Mom sit at chance everywhere. Llama-3.1-8B refuses every Fama-French query (parse rate 0 on all six factors), consistent with a capability floor below which the model declines to commit.

## C. Per-cell refusal / parse rates

Figure 4 and Table 5 report the fraction of queries that produced a committal answer (a parseable number for A/B; one of the two prompt months for C). Three patterns stand out: (i) Sonnet's Variant A parse rate tracks factor prominence (88% on Mkt-RF, $\sim$27% on RMW/CMA), consistent with self-knowledge of what it has memorized; (ii) Haiku's Variant B parse rate collapses on every factor ($\leq 31\%$); (iii) Haiku refuses essentially all of Variant C.

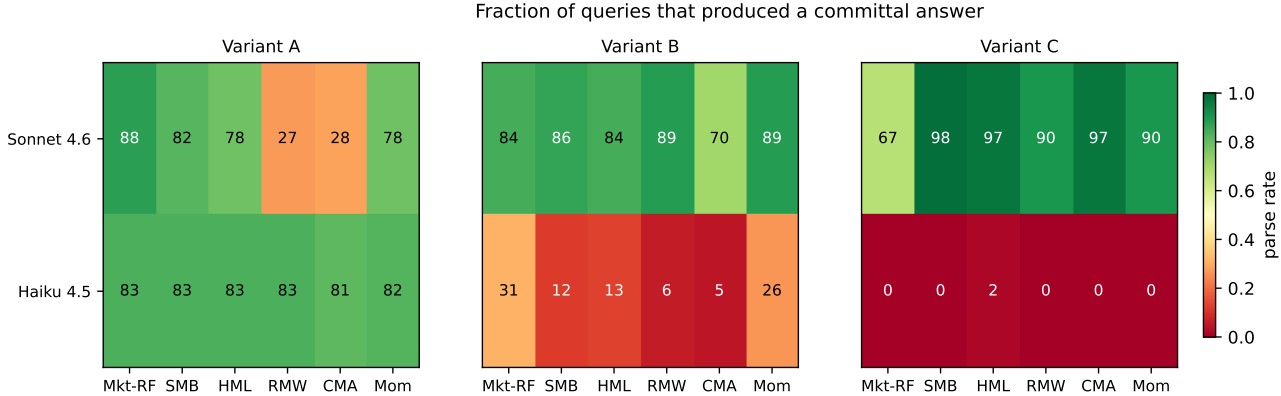

*Figure 4.* Parse rate per (model, factor, variant). Green = high commit rate; red = refusal. Values are percentages.

*Table 4.* Variant A headline metrics: nine frontier LLMs on the six Fama-French factors. Wilson-score 95% CIs on proportions; 1,000-sample bootstrap CI on Pearson $r$. "Sign" is conditional on non-zero truth. Bold: Mkt-RF rows. Mkt-RF $n$ comes from the 2,784-query main sweep for Sonnet/Haiku and from 40-month baseline probes for the other seven models; all other factors use 40-month probes. †Llama-3.1-8B refused every Fama-French query (parse rate $0/40$ per cell), so no statistic is computable; the empty-row pattern is itself the result.

| Model | Factor | $n$ | within-25 bps | Sign | Pearson $r$ |
|---|---|---|---|---|---|
| Opus 4.7 | **Mkt-RF** | 40 | **0.68** [0.52, 0.80] | **1.00** [0.91, 1.00] | **0.99** [0.97, 1.00] |
| Opus 4.7 | SMB | 40 | 0.12 [0.05, 0.26] | 0.78 [0.62, 0.88] | $+0.44$ [$-0.04$, 0.80] |
| Opus 4.7 | HML | 40 | 0.10 [0.04, 0.23] | 0.68 [0.52, 0.80] | $+0.58$ [$-0.29$, 0.91] |
| Opus 4.7 | RMW | 38 | 0.05 [0.01, 0.17] | 0.47 [0.32, 0.63] | $+0.16$ [$-0.44$, 0.70] |
| Opus 4.7 | CMA | 39 | 0.10 [0.04, 0.24] | 0.46 [0.32, 0.61] | $+0.12$ [$-0.48$, 0.64] |
| Opus 4.7 | Mom | 39 | 0.03 [0.00, 0.13] | 0.41 [0.27, 0.57] | $-0.35$ [$-0.80$, 0.16] |
| Sonnet 4.6 | **Mkt-RF** | 77 | **0.34** [0.24, 0.45] | **0.97** [0.91, 0.99] | **0.98** [0.96, 0.99] |
| Sonnet 4.6 | SMB | 72 | 0.08 [0.04, 0.17] | 0.61 [0.50, 0.72] | $-0.25$ [$-0.63$, 0.38] |
| Sonnet 4.6 | HML | 69 | 0.03 [0.01, 0.10] | 0.49 [0.38, 0.61] | $+0.48$ [0.15, 0.68] |
| Sonnet 4.6 | RMW | 24 | 0.04 [0.01, 0.20] | 0.54 [0.35, 0.72] | $+0.14$ [$-0.35$, 0.64] |
| Sonnet 4.6 | CMA | 25 | 0.00 [0.00, 0.13] | 0.60 [0.41, 0.77] | $+0.11$ [$-0.18$, 0.40] |
| Sonnet 4.6 | Mom | 69 | 0.01 [0.00, 0.08] | 0.48 [0.37, 0.59] | $-0.00$ [$-0.35$, 0.37] |
| Haiku 4.5 | **Mkt-RF** | 70 | **0.17** [0.10, 0.28] | **0.77** [0.66, 0.85] | **0.68** [0.51, 0.82] |
| Haiku 4.5 | SMB | 70 | 0.03 [0.01, 0.10] | 0.61 [0.50, 0.72] | $+0.45$ [0.24, 0.63] |
| Haiku 4.5 | HML | 70 | 0.10 [0.05, 0.19] | 0.64 [0.52, 0.74] | $+0.04$ [$-0.30$, 0.40] |
| Haiku 4.5 | RMW | 70 | 0.07 [0.03, 0.16] | 0.44 [0.33, 0.56] | $-0.34$ [$-0.51$, $-0.19$] |
| Haiku 4.5 | CMA | 68 | 0.10 [0.05, 0.20] | 0.50 [0.38, 0.62] | $-0.16$ [$-0.39$, 0.06] |
| Haiku 4.5 | Mom | 69 | 0.00 [0.00, 0.05] | 0.46 [0.35, 0.58] | $-0.25$ [$-0.55$, 0.10] |
| GPT-5.4 | **Mkt-RF** | 40 | **0.35** [0.22, 0.50] | **0.80** [0.65, 0.90] | **0.70** [0.42, 0.89] |
| GPT-5.4 | SMB | 40 | 0.10 [0.04, 0.23] | 0.70 [0.55, 0.82] | $-0.07$ [$-0.65$, 0.78] |
| GPT-5.4 | HML | 40 | 0.12 [0.05, 0.26] | 0.65 [0.50, 0.78] | $-0.06$ [$-0.65$, 0.71] |
| GPT-5.4 | RMW | 40 | 0.15 [0.07, 0.29] | 0.65 [0.50, 0.78] | $+0.28$ [$-0.50$, 0.81] |
| GPT-5.4 | CMA | 40 | 0.05 [0.01, 0.17] | 0.42 [0.29, 0.58] | $+0.27$ [$-0.45$, 0.80] |
| GPT-5.4 | Mom | 40 | 0.00 [0.00, 0.09] | 0.50 [0.35, 0.65] | $-0.03$ [$-0.55$, 0.29] |
| GPT-5.4-mini | **Mkt-RF** | 40 | **0.35** [0.22, 0.50] | **0.72** [0.57, 0.84] | **0.65** [0.32, 0.85] |
| GPT-5.4-mini | SMB | 40 | 0.10 [0.04, 0.23] | 0.50 [0.35, 0.65] | $+0.40$ [$-0.05$, 0.72] |
| GPT-5.4-mini | HML | 40 | 0.00 [0.00, 0.09] | 0.45 [0.31, 0.60] | $+0.01$ [$-0.41$, 0.35] |
| GPT-5.4-mini | RMW | 40 | 0.15 [0.07, 0.29] | 0.53 [0.37, 0.67] | $+0.13$ [$-0.28$, 0.51] |
| GPT-5.4-mini | CMA | 40 | 0.05 [0.01, 0.17] | 0.42 [0.29, 0.58] | $-0.25$ [$-0.54$, 0.05] |
| GPT-5.4-mini | Mom | 40 | 0.12 [0.05, 0.26] | 0.47 [0.33, 0.63] | $-0.02$ [$-0.48$, 0.47] |
| GPT-5.4-nano | **Mkt-RF** | 40 | **0.03** [0.00, 0.13] | **0.42** [0.29, 0.58] | **$-0.32$** [$-0.61$, 0.06] |
| GPT-5.4-nano | SMB | 40 | 0.07 [0.03, 0.20] | 0.42 [0.29, 0.58] | $-0.08$ [$-0.41$, 0.26] |
| GPT-5.4-nano | HML | 40 | 0.07 [0.03, 0.20] | 0.50 [0.35, 0.65] | $-0.09$ [$-0.42$, 0.27] |
| GPT-5.4-nano | RMW | 40 | 0.10 [0.04, 0.23] | 0.47 [0.33, 0.63] | $-0.27$ [$-0.55$, 0.02] |
| GPT-5.4-nano | CMA | 40 | 0.07 [0.03, 0.20] | 0.57 [0.42, 0.71] | $+0.26$ [$-0.17$, 0.56] |
| GPT-5.4-nano | Mom | 40 | 0.05 [0.01, 0.17] | 0.40 [0.26, 0.55] | $-0.08$ [$-0.35$, 0.19] |
| DeepSeek-V3.2 | **Mkt-RF** | 40 | **0.15** [0.07, 0.29] | **0.72** [0.57, 0.84] | **0.48** [0.15, 0.73] |
| DeepSeek-V3.2 | SMB | 40 | 0.05 [0.01, 0.17] | 0.70 [0.55, 0.82] | $+0.46$ [$+0.05$, 0.71] |
| DeepSeek-V3.2 | HML | 40 | 0.03 [0.00, 0.13] | 0.40 [0.26, 0.55] | $-0.06$ [$-0.37$, 0.30] |
| DeepSeek-V3.2 | RMW | 40 | 0.05 [0.01, 0.17] | 0.42 [0.29, 0.58] | $+0.07$ [$-0.23$, 0.43] |
| DeepSeek-V3.2 | CMA | 40 | 0.07 [0.03, 0.20] | 0.47 [0.33, 0.63] | $-0.16$ [$-0.51$, 0.19] |
| DeepSeek-V3.2 | Mom | 40 | 0.03 [0.00, 0.13] | 0.38 [0.24, 0.53] | $-0.30$ [$-0.62$, $-0.16$] |
| Llama-3.3-70B | **Mkt-RF** | 39 | **0.08** [0.03, 0.20] | **0.62** [0.46, 0.75] | **0.31** [$-0.09$, 0.60] |
| Llama-3.3-70B | SMB | 40 | 0.05 [0.01, 0.17] | 0.65 [0.50, 0.78] | $-0.08$ [$-0.36$, 0.20] |
| Llama-3.3-70B | HML | 40 | 0.00 [0.00, 0.09] | 0.45 [0.31, 0.60] | $+0.08$ [$-0.41$, 0.57] |
| Llama-3.3-70B | RMW | 40 | 0.00 [0.00, 0.09] | 0.42 [0.29, 0.58] | $-0.02$ [$-0.47$, 0.41] |
| Llama-3.3-70B | CMA | 40 | 0.12 [0.05, 0.26] | 0.47 [0.33, 0.63] | $+0.21$ [$+0.03$, 0.40] |
| Llama-3.3-70B | Mom | 40 | 0.05 [0.01, 0.17] | 0.42 [0.29, 0.58] | $-0.26$ [$-0.50$, $-0.02$] |
| Llama-3.1-8B† | **Mkt-RF** | 40 | **–** | **–** | **–** |
| Llama-3.1-8B† | SMB | 40 | – | – | – |
| Llama-3.1-8B† | HML | 40 | – | – | – |
| Llama-3.1-8B† | RMW | 40 | – | – | – |
| Llama-3.1-8B† | CMA | 40 | – | – | – |
| Llama-3.1-8B† | Mom | 40 | – | – | – |

*Table 5.* Parse rate (fraction of queries with a committal answer). Denominator $n$ is queries that completed without an API error.

| Model | Factor | Variant A | Variant B | Variant C |
|-------|--------|-----------|-----------|-----------|
| Sonnet 4.6 | Mkt-RF | 0.88 | 0.84 | 0.67 |
| Sonnet 4.6 | SMB | 0.82 | 0.86 | 0.98 |
| Sonnet 4.6 | HML | 0.78 | 0.84 | 0.97 |
| Sonnet 4.6 | RMW | 0.27 | 0.89 | 0.90 |
| Sonnet 4.6 | CMA | 0.28 | 0.70 | 0.97 |
| Sonnet 4.6 | Mom | 0.78 | 0.89 | 0.90 |
| Haiku 4.5 | Mkt-RF | 0.83 | 0.31 | 0.00 |
| Haiku 4.5 | SMB | 0.83 | 0.12 | 0.00 |
| Haiku 4.5 | HML | 0.83 | 0.13 | 0.02 |
| Haiku 4.5 | RMW | 0.83 | 0.06 | 0.00 |
| Haiku 4.5 | CMA | 0.81 | 0.05 | 0.00 |
| Haiku 4.5 | Mom | 0.82 | 0.26 | 0.00 |

## D. Per-cell cutoff-gradient regressions

Table 6 reports the per-cell OLS of within-25 bps recall on signed months-to-cutoff, pooling Variants A and B. Raw two-sided $p$-values are computed against a $t$-distribution with $n_{\text{bins}}-2$ degrees of freedom; $q$-values are Benjamini–Hochberg-adjusted across all 12 tests at $\alpha=0.05$.

Two Mom-like cells (Haiku Mom, Sonnet CMA) have zero variance in the outcome (all months miss the 25 bps threshold), so the OLS is degenerate: slope $= 0$, stderr $= 0$, and the t-statistic is undefined. We report them as "–" since a null-variance regression cannot support or reject a gradient. No surviving cell, including Haiku×Mkt-RF (raw $p=0.030$, BH-$q=0.12$), clears FDR at $q=0.05$.

*Table 6.* Per-cell cutoff-gradient OLS, Variants A∪B. Slope is in recall-rate per month; positive = recall higher farther inside training data.

| Model | Factor | $n_{\text{bins}}$ | slope | stderr | $r^2$ | raw $p$ | BH-$q$ |
|-------|--------|------|-------|--------|-------|---------|--------|
| Sonnet 4.6 | Mkt-RF | 83 | $-3.0 \times 10^{-4}$ | $2.0 \times 10^{-4}$ | 0.027 | 0.135 | 0.41 |
| Sonnet 4.6 | SMB | 82 | $-3 \times 10^{-6}$ | $1.4 \times 10^{-4}$ | 0.000 | 0.852 | 0.86 |
| Sonnet 4.6 | HML | 80 | $-7 \times 10^{-6}$ | $9 \times 10^{-5}$ | 0.007 | 0.452 | 0.73 |
| Sonnet 4.6 | RMW | 80 | $-2 \times 10^{-5}$ | $1.1 \times 10^{-4}$ | 0.000 | 0.861 | 0.86 |
| Sonnet 4.6 | CMA | 65 | – | – | – | – | – |
| Sonnet 4.6 | Mom | 79 | $+4 \times 10^{-6}$ | $9 \times 10^{-5}$ | 0.003 | 0.654 | 0.86 |
| Haiku 4.5 | Mkt-RF | 70 | $-4.4 \times 10^{-4}$ | $2.0 \times 10^{-4}$ | 0.067 | 0.030 | 0.12 |
| Haiku 4.5 | SMB | 70 | $+3 \times 10^{-5}$ | $1.5 \times 10^{-4}$ | 0.001 | 0.847 | 0.86 |
| Haiku 4.5 | HML | 71 | $+1.7 \times 10^{-4}$ | $1.8 \times 10^{-4}$ | 0.013 | 0.352 | 0.73 |
| Haiku 4.5 | RMW | 71 | $+9 \times 10^{-5}$ | $1.0 \times 10^{-4}$ | 0.011 | 0.380 | 0.73 |
| Haiku 4.5 | CMA | 68 | $+1.8 \times 10^{-4}$ | $2.5 \times 10^{-4}$ | 0.007 | 0.490 | 0.73 |
| Haiku 4.5 | Mom | 69 | – | – | – | – | – |

## E. Pre-vs-post cutoff stratification

The temporal-nulls analysis (§3.1) reports slope-based tests on (recall vs. months-to-cutoff). To complement these slope tests with a discrete pre/post split, we bucket each probed Mkt-RF Variant-A month relative to its model's training cutoff: *pre* ($d>6$ months before cutoff), *near* ($|d|\leq6$), and *post* ($d>6$ months after).

*Table 7.* Mkt-RF Variant-A recall stratified by months-to-cutoff bucket. "parse" is parsed/sampled. $r$ and within-25 bps are on the parsed subset; "–" marks cells where the parsed subset is too small ($n<3$) for a correlation. Anthropic Sonnet/Haiku include the multi-seed pool; the other six rows are 40-month baseline probes whose sampling drew 0 post-cutoff months by construction (uninformative for stratification, reported for transparency).

| Model | Cutoff | Pre | | | Near | | | Post | | |
|---|---|---|---|---|---|---|---|---|---|---|
| | | parse | $r$ | w25 | parse | $r$ | w25 | parse | $r$ | w25 |
| Sonnet 4.6 | 2025-03 | 70/70 | +0.98 | 0.36 | 7/13 | +0.99 | 0.14 | 0/5 | – | – |
| Haiku 4.5 | 2025-07 | 70/70 | +0.68 | 0.19 | 0/13 | – | – | 0/1 | – | – |
| Opus 4.7 | 2026-01 | 40/40 | +0.99 | 0.68 | 0/0 | – | – | 0/0 | – | – |
| GPT-5.4 | 2025-08 | 40/40 | +0.70 | 0.35 | 0/0 | – | – | 0/0 | – | – |
| GPT-5.4-mini | 2025-08 | 40/40 | +0.65 | 0.35 | 0/0 | – | – | 0/0 | – | – |
| GPT-5.4-nano | 2025-08 | 40/40 | −0.32 | 0.03 | 0/0 | – | – | 0/0 | – | – |
| DeepSeek-V3.2 | 2025-07 | 40/40 | +0.48 | 0.15 | 0/0 | – | – | 0/0 | – | – |
| Llama-3.3-70B | 2024-12 | 39/39 | +0.31 | 0.08 | 0/1 | – | – | 0/0 | – | – |

**Two findings.** (i) *Refusal cliff at the cutoff.* Sonnet × Mkt-RF parses 70/70 pre-cutoff, 7/13 near, and 0/5 post; Haiku parses 70/70 pre, 0/13 near, 0/1 post. Anthropic models that produce committal answers pre-cutoff sharply suppress their own outputs on post-cutoff months, consistent with self-knowledge of training cutoff. The post-cutoff $n$ is small by design: the probe window ends 2026-02 and Anthropic mid-2025 cutoffs leave $\sim 8$ months past-cutoff, of which the per-cell sampler hits the few that survive truth-availability gates. (ii) *Pre-cutoff recall fidelity is unchanged across buckets.* On the parsed subset, Sonnet $r=0.980$ pre and $0.989$ near are statistically indistinguishable (bootstrap CIs overlap). The eight 40-month baseline rows draw 0 post-cutoff months by construction, so the stratification is uninformative for those cells.

**Reconciliation with the slope-based null.** §3.1 reports that the within-25 bps *rate* does not gradient with months-to-cutoff. The stratified analysis sharpens this: among parsed responses, the rate is flat in cutoff distance, but *parsing itself* drops sharply across the cutoff boundary on Sonnet and Haiku. The cutoff effect is therefore discrete (refusal/no-refusal), not a gradient on recall fidelity. The slope-based test, which ignores refusal, misses this dimension; we now read the two together as "training-cutoff awareness manifests as refusal, recall fidelity is uniform among answered months."

## F. Baselines and label invariance

Three auxiliary probes characterize *what* Sonnet has memorized: an S&P 500 probe, a NASDAQ Composite probe, and a blind-label probe that asks for "the broad U.S. stock market in excess of the T-bill rate" without naming Fama-French. Truth for S&P 500 and NASDAQ comes from Yahoo Finance monthly close-to-close price returns; truth for the blind probe is Kenneth French Mkt-RF. Table 8 reports recall on the same Variant-A answer format across all three alongside the main-sweep Mkt-RF row.

*Table 8.* Cross-model recall on four probes for the aggregate U.S. equity return. $\rho_{FF}$ is the correlation of the target truth series with Ken French Mkt-RF on the probed months. $n{=}40$ per cell for the baselines; the Sonnet main-sweep Mkt-RF row uses $n{=}77$. Anthropic models, three OpenAI GPT-5.4 tiers, DeepSeek-V3.2, and the two Meta Llamas, all via official APIs. Llama-3.1-8B refuses every Mkt-RF query (parse rate 0); $r$ is reported on the parsed subset. GPT-5.4-nano's Mkt-RF row is the only negative $r$ in the table; the smallest GPT model generates anti-correlated noise rather than the memorized series.

| Model | Probe | $\rho_{FF}$ | parse | within-25 bps | Pearson $r$ | sign |
|---|---|---|---|---|---|---|
| Opus 4.7 | Mkt-RF | 1.00 | 1.00 | 0.68 | +0.986 | 1.00 |
| Opus 4.7 | S&P 500 | 0.99 | 1.00 | 1.00 | +1.000 | 1.00 |
| Opus 4.7 | NASDAQ Composite | 0.92 | 1.00 | 0.88 | +0.972 | 0.93 |
| Opus 4.7 | Blind U.S. mkt excess | 1.00 | 1.00 | 0.68 | +0.954 | 0.98 |
| Sonnet 4.6 | Mkt-RF (main) | 1.00 | 0.88 | 0.34 | +0.98 | 0.97 |
| Sonnet 4.6 | S&P 500 | 0.99 | 1.00 | 0.85 | +0.97 | 0.95 |
| Sonnet 4.6 | NASDAQ Composite | 0.92 | 0.95 | 0.63 | +0.81 | 0.84 |
| Sonnet 4.6 | Blind U.S. mkt excess | 1.00 | 0.62 | 0.20 | +0.92 | 1.00 |
| Haiku 4.5 | S&P 500 | 0.99 | 1.00 | 0.38 | +0.59 | 0.75 |
| Haiku 4.5 | NASDAQ Composite | 0.92 | 0.93 | 0.08 | +0.48 | 0.76 |
| GPT-5.4 | Mkt-RF | 1.00 | 1.00 | 0.35 | +0.70 | 0.80 |
| GPT-5.4 | S&P 500 | 0.99 | 1.00 | 0.63 | +0.91 | 0.88 |
| GPT-5.4 | NASDAQ Composite | 0.92 | 1.00 | 0.23 | +0.71 | 0.78 |
| GPT-5.4 | Blind U.S. mkt excess | 1.00 | 1.00 | 0.33 | +0.77 | 0.85 |
| GPT-5.4-mini | Mkt-RF | 1.00 | 1.00 | 0.35 | +0.65 | 0.73 |
| GPT-5.4-mini | S&P 500 | 0.99 | 1.00 | 0.50 | +0.76 | 0.83 |
| GPT-5.4-mini | NASDAQ Composite | 0.92 | 1.00 | 0.15 | +0.43 | 0.70 |
| GPT-5.4-mini | Blind U.S. mkt excess | 1.00 | 1.00 | 0.10 | +0.54 | 0.70 |
| GPT-5.4-nano | Mkt-RF | 1.00 | 1.00 | 0.03 | −0.32 | 0.43 |
| GPT-5.4-nano | S&P 500 | 0.99 | 1.00 | 0.08 | +0.43 | 0.60 |
| GPT-5.4-nano | NASDAQ Composite | 0.92 | 1.00 | 0.10 | +0.20 | 0.50 |
| GPT-5.4-nano | Blind U.S. mkt excess | 1.00 | 1.00 | 0.05 | +0.18 | 0.65 |
| DeepSeek-V3.2 | Mkt-RF | 1.00 | 1.00 | 0.15 | +0.48 | 0.73 |
| DeepSeek-V3.2 | S&P 500 | 0.99 | 1.00 | 0.55 | +0.86 | 0.83 |
| DeepSeek-V3.2 | NASDAQ Composite | 0.92 | 1.00 | 0.23 | +0.80 | 0.73 |
| DeepSeek-V3.2 | Blind U.S. mkt excess | 1.00 | 1.00 | 0.15 | +0.42 | 0.65 |
| Llama-3.3-70B | Mkt-RF | 1.00 | 0.97 | 0.08 | +0.31 | 0.62 |
| Llama-3.3-70B | S&P 500 | 0.99 | 1.00 | 0.45 | +0.68 | 0.65 |
| Llama-3.3-70B | NASDAQ Composite | 0.92 | 1.00 | 0.10 | +0.18 | 0.60 |
| Llama-3.3-70B | Blind U.S. mkt excess | 1.00 | 1.00 | 0.10 | +0.08 | 0.60 |
| Llama-3.1-8B | Mkt-RF | 1.00 | 0.00 | – | – | – |
| Llama-3.1-8B | S&P 500 | 0.99 | 1.00 | 0.03 | +0.23 | 0.40 |
| Llama-3.1-8B | NASDAQ Composite | 0.92 | 0.55 | 0.00 | −0.03 | 0.50 |
| Llama-3.1-8B | Blind U.S. mkt excess | 1.00 | 0.53 | 0.00 | +0.13 | 0.33 |

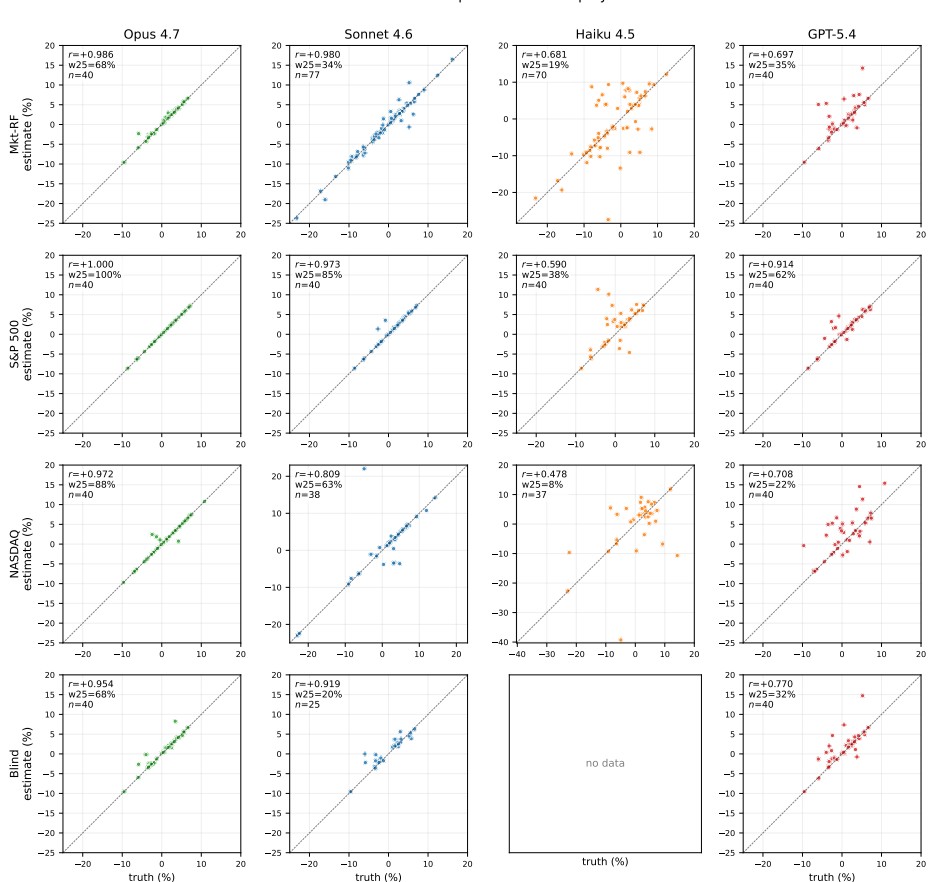

*Figure 5.* Calibration scatter for every (model, probe) cell of the *original four* models in Table 8. Rows are probes (Mkt-RF, S&P 500, NASDAQ, blind); columns are models (Opus, Sonnet, Haiku, GPT-5.4). Per-cell annotations: Pearson $r$, within-25 bps rate, and parsed $n$. Haiku's blind-probe cell is empty because we did not probe Haiku blind. The five additional models in Table 8 (GPT-5.4-mini/nano, DeepSeek-V3.2, Llama-3.3-70B, Llama-3.1-8B) are summarized in Fig. 6.

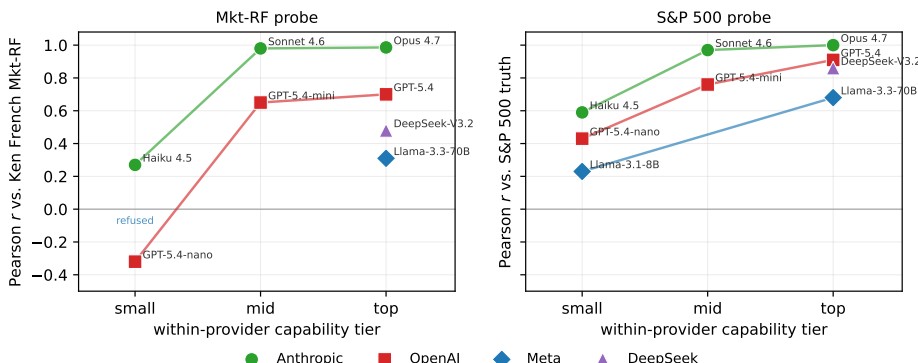

*Figure 6.* Capability-scaled recall across providers. Recall increases with within-provider model tier on Mkt-RF and S&P 500; DeepSeek provides an additional non-U.S. provider check.

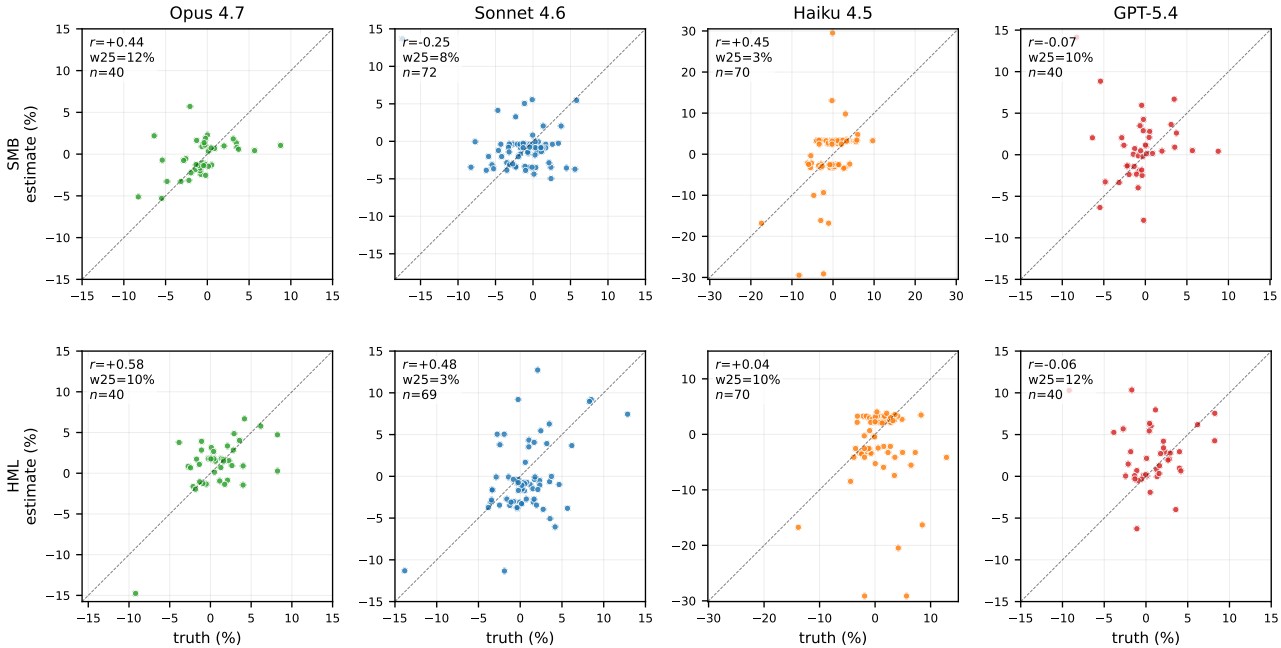

*Figure 7.* Variant-A calibration on the two Fama-French factors with any partial recall (SMB, HML) across all four models. Opus shows the cleanest alignment ($r$=0.44 on SMB, $r$=0.58 on HML), with weaker but visible HML signal on Sonnet ($r$=0.48); other cells are noise. Mkt-RF (clean recall on all four) is shown in Fig. 2 and the top row of Fig. 5; RMW, CMA, and Mom are at chance on every model and not shown.

## G. Non-LLM proxy and shifted-date controls

The direct recall results establish a high-correlation interface to Mkt-RF, but correlation alone is not sufficient to identify the literal Kenneth French normalization. We therefore compare the model results with non-LLM baselines evaluated on the same month sets. The calendar and event baselines use only information available before the queried month: an expanding historical mean, an expanding mean for the same calendar month, and an expanding mean over the famous-month list when the queried month is famous. The same-series shuffle pairs each queried month with another queried month's Mkt-RF. The S&P 500 and NASDAQ baselines use expanding OLS regressions from each index's monthly return to Kenneth French Mkt-RF, trained only on months before the queried month.

*Table 9.* Non-LLM baselines on the same Mkt-RF month sets used for Sonnet and Opus. Calendar, event, and shuffle baselines are weak. An S&P 500 proxy is highly correlated with Mkt-RF, showing that correlation alone cannot distinguish exact Fama–French recall from recall or reconstruction of a correlated aggregate-market index. Proxy rows use only months with index-data coverage (S&P 500/NASDAQ truth begins after Kenneth French Mkt-RF), so $n$ is smaller and the comparison is not directly sample-identical for the earliest sampled months. Scripts: `experiments/54_proxy_calendar_baselines.py`.

| Month set | Baseline | $n$ | $r$ | w-25 bps | MAE |
|---|---|---|---|---|---|
| Sonnet | expanding mean | 88 | +0.03 | 0.09 | 4.49 |
| Sonnet | famous-month prior | 88 | +0.39 | 0.11 | 3.97 |
| Sonnet | same-series shuffle | 88 | +0.09 | 0.02 | 6.43 |
| Sonnet | NASDAQ proxy | 74 | +0.93 | 0.10 | 1.71 |
| Sonnet | S&P 500 proxy | 74 | +0.99 | 0.38 | 0.55 |
| Opus | expanding mean | 40 | +0.06 | 0.08 | 2.99 |
| Opus | famous-month prior | 40 | +0.44 | 0.08 | 2.81 |
| Opus | same-series shuffle | 40 | +0.23 | 0.10 | 3.57 |
| Opus | NASDAQ proxy | 20 | +0.85 | 0.05 | 1.54 |
| Opus | S&P 500 proxy | 20 | +0.99 | 0.50 | 0.40 |

To test whether the value readout follows the prompted date rather than an artifact of the sampled month set, we ran a shifted-date control on Sonnet and Opus. For each of 30 original Mkt-RF months per model, we queried Variant A after

shifting the prompt by $+12$ months, $-12$ months, or by six months within the same year. The model output was then compared both to the prompted date's Mkt-RF and to the original sampled date's Mkt-RF.

*Table 10.* Shifted-date control for Mkt-RF. Values track the prompted date, not the original sampled date. The run used 180 API calls and spent \$0.2216 under a \$5 cap. Script: `experiments/55_date_shift_control.py`.

| Model | Shift | $n$ | $r_{\text{prompted}}$ | $r_{\text{original}}$ | w-25 bps$_{\text{prompted}}$ |
|---|---|---|---|---|---|
| Opus | $-12$ months | 30 | $+0.984$ | $+0.094$ | 0.633 |
| Opus | $+12$ months | 30 | $+0.993$ | $+0.071$ | 0.600 |
| Opus | within-year swap | 30 | $+0.973$ | $+0.079$ | 0.633 |
| Sonnet | $-12$ months | 29 | $+0.949$ | $-0.116$ | 0.345 |
| Sonnet | $+12$ months | 29 | $+0.860$ | $-0.326$ | 0.276 |
| Sonnet | within-year swap | 29 | $+0.962$ | $-0.182$ | 0.310 |

Together, these controls sharpen the estimand. Calendar/event priors and same-series shuffles cannot reproduce Mkt-RF recall. A correlated S&P 500 proxy can reproduce much of the correlation, so the strongest claim is not literal-label exclusivity. The shifted-date control shows that the ordinary query interface follows the prompted date, which is the behavior required for date-indexed benchmark labels to become accessible through model outputs.

## H. Multi-seed robustness (Mkt-RF)

Table 11 reports per-seed recall on 40 random Mkt-RF months for seeds $\{1, 2, 3\}$, plus the pooled statistics across the three runs. Sonnet's pooled $r=0.92$ is consistent with the main-sweep value. Haiku's pooled $r=0.27$ is much lower than the main-sweep $r=0.68$: the single-seed result was a favorable draw. The pooled value ($r=0.27$) is the preferred point estimate; the main panel (Tab. 1) does not include Haiku, but pooled values are used in any appendix reference to Haiku Mkt-RF recall.

*Table 11.* Per-seed and pooled Mkt-RF recall under Variant A. Main-sweep (seed 42) rows: Sonnet $r=0.98$, within-25 bps=0.338, sign $= 0.974$; Haiku $r=0.68$, within-25 bps=0.171, sign $= 0.771$.

| Model | Seed | $n$ | Pearson $r$ | within-25 bps | sign |
|---|---|---|---|---|---|
| Sonnet 4.6 | 1 | 39 | $+0.858$ | 0.359 | 0.923 |
| Sonnet 4.6 | 2 | 40 | $+0.967$ | 0.175 | 0.925 |
| Sonnet 4.6 | 3 | 40 | $+0.953$ | 0.250 | 0.950 |
| Sonnet 4.6 | pooled | 119 | $+0.921$ | 0.261 | 0.933 |
| Haiku 4.5 | 1 | 40 | $+0.586$ | 0.175 | 0.700 |
| Haiku 4.5 | 2 | 40 | $+0.021$ | 0.125 | 0.625 |
| Haiku 4.5 | 3 | 40 | $+0.287$ | 0.050 | 0.625 |
| Haiku 4.5 | pooled | 120 | $+0.266$ | 0.117 | 0.650 |

## I. Cross-domain replication: U.S. unemployment rate

To address the concern that series memorization may be specific to Fama-French, we replicate the headline Variant-A probe on the Bureau of Labor Statistics monthly civilian unemployment rate (FRED series `UNRATE`, seasonally adjusted), a different domain (macro/labor), different canonical source (BLS, not Ken French), and different sign convention (always-positive level). We sample 30 months from 1980–2024 (seed 42) and ask each model for a single-decimal percent.

| Model | $n$ | parse | $r$ | within-25bps | within-50bps |
|---|---|---|---|---|---|
| Sonnet 4.6 | 30 | 1.00 | $+1.000$ | 1.00 | 1.00 |
| Opus 4.7 | 30 | 1.00 | $+1.000$ | 1.00 | 1.00 |

*Table 12.* UNRATE recall on Sonnet/Opus: every one of 60 monthly queries produces an exact-decimal answer matching the BLS-published value within 0.25 percentage points. Note that UNRATE has $\sigma \approx 0.1$ pp/month (vs. Mkt-RF $\sigma \approx 4.5\%$/month), so the within-25 bps tolerance is a much weaker test of fidelity than on Mkt-RF; the result demonstrates the identification framework is *domain-portable*, not that UNRATE is recalled at higher fidelity than Mkt-RF.

## J. Cross-domain replication: CPI YoY inflation

A second non-financial replication on a different macro category (price level, not labor): U.S. year-over-year CPI inflation rate (FRED `CPIAUCSL`, computed as 12-month percent change from the level series). 30 months sampled from 1980–2024 (seed 2028, script `experiments/50_cpi_baseline.py`).

| Model | $n$ | parse | $r$ | within-25 bps |
|---|---|---|---|---|
| Sonnet 4.6 | 30 | 0.97 | +0.995 | 0.93 |
| Opus 4.7 | 30 | 1.00 | +1.000 | 1.00 |

*Table 13.* CPI YoY recall on Sonnet/Opus. CPI YoY has higher month-to-month variance than UNRATE (range $-2$ to $14\%$ across the sample) so the within-25 bps test is a stronger fidelity check here. Two non-financial series across distinct macro categories (labor + prices) both recall above $r=0.99$ on the top tier; the identification framework is domain-portable across more than just UNRATE.

## K. Auxiliary probes: variants C/D/E

Three auxiliary probes reveal the structure of what is memorized. **Variant C (comparative)**: Haiku refuses $99.7\%$ of 360 pairs; Sonnet answers $89.7\%$ across all six factors. On Sonnet×Mkt-RF specifically ($n=60$ pairs, where values are recalled at $r=0.98$) rank accuracy is at chance under three independent measurements (Tab. 14): endorsement-aware parser (App. K.1) on the parsed subset gives $52.5\%$ (parse 40/60); a naive "first month mentioned" parser at near-full parse gives $49.2\%$ ($n=59$); and a forced-choice rerun with a strict prompt that drives parse to $100\%$ gives $55.0\%$ ($n=60$). All three $95\%$ binomial CIs include $50\%$, so the chance-level result is robust both to parser choice and to refusal-based selection bias.

### K.1. Variant-C parser algorithm

The comparative parser is endorsement-aware: it must handle preambles that echo the prompt ("Between March 2020 and October 2008, ...") before the model commits. Pseudocode:

1. Normalize Unicode minus / en-dash / hyphen variants to ASCII.

2. Collect every (`offset, YYYY-MM`) mention in the response.

3. *Candidate filter.* Drop mentions that are not one of the two prompt months.

4. *Refusal guard.* If the text matches a refusal phrase ("I don't have access", "cannot reliably recall", "outside my training", ...) **and** contains no strong-endorse phrase, return `None`.

5. *Unique mention.* If only one candidate appears, return it.

6. *Strong endorsement.* If a phrase (*my answer is X, answer: X, I pick X, higher return was in X, ...*) names a candidate, return it.

7. *Prompt echo.* If the response opens with *Between, Comparing, The two months, ...*, the first mention is an echo; return the last candidate mention.

8. *Leading answer.* If the first candidate mention starts within the first 5 characters of the stripped response, return it (handles "March 2020 was higher.").

9. Otherwise, return the last candidate mention.

Refusals parsed under step 4 are excluded from the comparative-accuracy denominator, not counted as wrong picks; this is why Haiku's headline Variant-C accuracy is reported over a denominator of order 1, not 360.

| Measurement | parse | accuracy | 95% CI |
|---|---|---|---|
| Endorsement-aware (paper) | 0.67 | 0.525 | [0.370, 0.680] |
| Naive first-mention parser | 0.98 | 0.492 | [0.364, 0.619] |
| Forced-choice rerun | 1.00 | 0.550 | [0.424, 0.676] |

*Table 14.* Rank accuracy on Sonnet×Mkt-RF Variant-C pairs ($n$=60 unique pairs) under three measurement vari­ants. Forced choice uses a strict prompt requiring the model to commit to one of two month strings (script `experiments/47_variantc_forced_choice.py`); naive parser ignores refusal phrases and returns the first candidate month mentioned (script `experiments/48_variantc_parser_ablation.py`). All three 95% CIs include 50%.

**Variant-C extension to SMB and HML.** The decoupling claim above was Sonnet×Mkt-RF specific. We re-ran the endorsement-aware and naive-first-mention parsers on the existing sweep records for Sonnet×SMB ($n$=60, value recall $r$=−0.25) and Sonnet×HML ($n$=60, value recall $r$=+0.48) pairs (Tab. 15). On SMB both parsers give chance-level rank accuracy (47.5% and 41.7%, both 95% CIs include 50%), consistent with poor value recall. On HML the two parsers *disagree*: endorsement-aware gives 65.5% (CI [53.3%, 77.7%], above chance), while the naive parser gives 39.0% (below chance); the gap reflects that on partial-recall pairs the model's endorsed pick carries genuine signal that the naive parser discards as prompt echo. The *regime* pattern is therefore: on the high-recall factor (Mkt-RF, $r$=0.98) point values and pairwise ranks separate under the tested prompts; on partial recall (HML, $r$=0.48) ranks and values track together; on a factor with no useful positive value recall (SMB, $r$=−0.25) ranks are at chance. The value/rank split is clearest precisely where point recall is strongest.

| Factor (value recall) | Endorse-aware | Naive | $n$ |
|---|---|---|---|
| Mkt-RF ($r$=0.98) | 0.525 [0.37, 0.68] | 0.492 [0.36, 0.62] | 60 |
| HML ($r$=0.48) | 0.655 [0.53, 0.78] | 0.390 [0.27, 0.51] | 60 |
| SMB ($r$=−0.25) | 0.475 [0.35, 0.60] | 0.417 [0.29, 0.54] | 60 |

*Table 15.* Variant-C rank accuracy on Sonnet across three factors, both parsers (script `experiments/48_variantc_parser_ablation.py`; data from the existing main sweep). Mkt-RF and SMB are at chance under both parsers; on HML the parsers disagree, reflecting partial value recall that the endorsement-aware parser correctly attributes to the model's pick.

**Variant D (chain-of-thought)**: prepending "Think step-by-step" *reduces* recall sharply on Sonnet × Mkt-RF ($r$: 0.98→0.78, within-25 bps: 33.8%→14.9%; $n$=121). **Variant E** ($T$=1): accuracy essentially unchanged ($r$=0.983, within-25 bps 37.5%); two independent draws at the same month agree within 25 bps in 93% of pairs (mean spread 6 bps). The pattern is consistent with a direct value readout that does not automatically compose into reliable pairwise comparison under these prompts. We do not infer an indexable map, or the absence of one, from this behavioral result; the evidence establishes that accurate point recall can coexist with unreliable comparative answers. The CoT condition is therefore best interpreted as a probe perturbation that reduces recall in this setting, not as a general mitigation claim.

**Numerical detail.** Variant D (CoT) probes 133 Mkt-RF months at `max_tokens`= 384; Variant E (T= 1) probes 88 Mkt-RF months with two independent draws each (176 responses). Per-variant recall is summarized in Tab. 16. Figure 8 shows the paired degradation under CoT on the month-matched subset: Variant A's $r$=0.98 collapses to Variant D's $r$=0.82, and on 54 of 73 paired months the CoT absolute error is strictly larger than the direct error. For Variant E, the within-draw spread on the 75 months where both draws parsed is 6.3 bps on average; 93.3% of same-month pairs agree within 25 bps. Temperature does not disturb the committal readout; reasoning tokens do.

### Chain-of-thought degrades recall: Variant A (direct) vs Variant D (CoT)

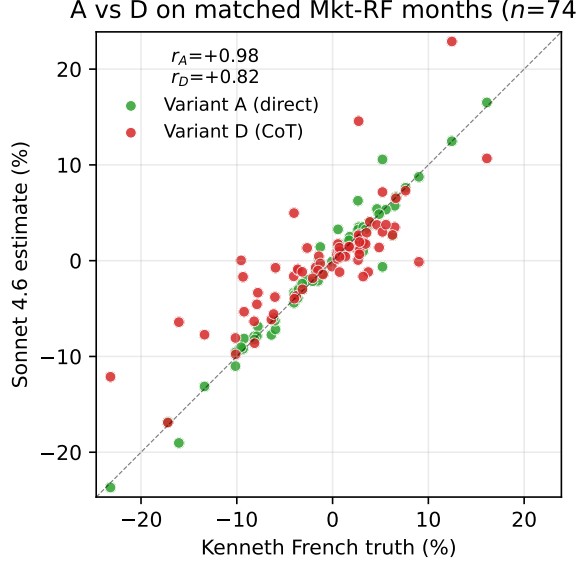
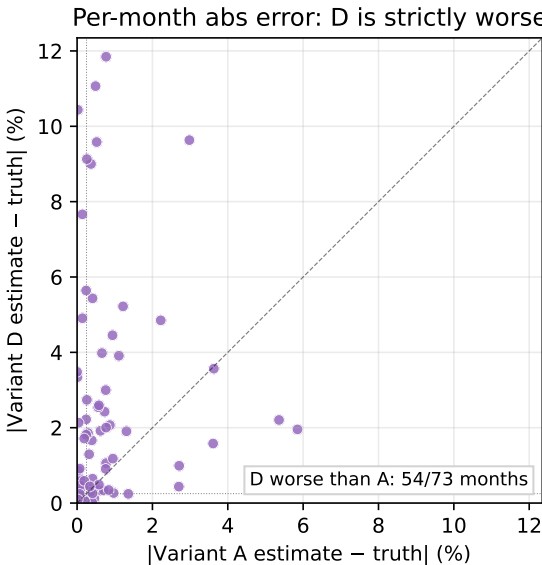

*Figure 8.* Chain-of-thought degrades Sonnet's Mkt-RF recall. *Left*: Variant A (green) and Variant D (red) estimates plotted against Kenneth French truth on the months probed under both conditions. *Right*: per-month absolute error; Variant D (y-axis) versus Variant A (x-axis). Points above the dashed equality line are months where reasoning made the answer worse.

*Table 16.* Mkt-RF recall under Variant D (CoT) and E (T= 1). Main-sweep Variant A on Sonnet for comparison: within-25 bps=0.338, $r$=0.980.

| Model | Variant | $n$ | parse rate | within-25 bps | Pearson $r$ |
|---|---|---|---|---|---|
| Sonnet 4.6 | D (CoT) | 133 | 0.910 | 0.149 | +0.776 |
| Haiku 4.5 | D (CoT) | 133 | 0.602 | 0.100 | +0.702 |
| Sonnet 4.6 | E (T= 1) | 176 | 1.000 | 0.382 | +0.983 |

## L. Phrasing-perturbation control

The Variant-A template fixes a specific surface form ("*What was the monthly return of the Fama-French [factor] factor in [month]? Answer with a signed decimal percentage . . .*"). This control separates template sensitivity from parametric representation of the series. We hold content fixed (Mkt-RF monthly return, signed-percent answer) and vary only the surface form across three reworded prompts:

1. **Terse**: "*Mkt-RF (Fama-French market excess) monthly return in {month}?*"

2. **Imperative**: "*Tell me the Fama-French Mkt-RF return for {month}.*"

3. **Conversational**: "*Do you remember what the Fama-French Mkt-RF factor returned in {month}?*"

Sample: 30 months from 1980–2020 (seed 2027, fresh draw) on Sonnet 4.6 and Opus 4.7 (script `experiments/49_phrasing_perturbation.py`).

| Model | Phrasing | $n$ | parse | $r$ | w-25 bps |
|---|---|---|---|---|---|
| Opus 4.7 | terse | 30 | 1.00 | +0.937 | 0.50 |
| Opus 4.7 | imperative | 30 | 1.00 | +0.950 | 0.63 |
| Opus 4.7 | conversational | 30 | 1.00 | +0.978 | 0.57 |
| Sonnet 4.6 | terse | 30 | 1.00 | +0.926 | 0.33 |
| Sonnet 4.6 | imperative | 30 | 1.00 | +0.911 | 0.23 |
| Sonnet 4.6 | conversational | 30 | 1.00 | +0.973 | 0.40 |

*Table 17.* Mkt-RF recall under three reworded Variant-A prompts. Main-sweep baseline (Variant A): Sonnet $r=0.98$ ($n=77$), Opus $r=0.99$ ($n=40$). All six (model, phrasing) cells parse 100% and recover $r \geq 0.91$. Within-25 bps rates drop modestly under terser/imperative phrasings (more one-significant-figure rounding), but rank-correlation $r$ is stable. The recall channel is *not* a property of the exact template wording.

## M. Expanded fabricated-series control

The original fabricated-series probe used two fictional names on Sonnet/Haiku ($n=24$) and was acknowledged in the main text as underpowered. We expand to five fictional names × eight models × twelve months ($n=480$ over four providers, seed 2026, script `experiments/46_fabricated_expansion.py`). The prompt is identical to Variant A except the factor name is replaced by one of: *Gleason-Zeta volatility-conditioned residual factor*, *Holbrooke-Mansfield Opportunity Fund III (2007 vintage)*, *Brennan-Iyer mean-reversion premium factor*, *Northrop-Calloway long-horizon dispersion factor*, *Pemberton-Yi cross-sectional liquidity premium factor*. These constructed names are treated as unsupported-series controls.

| Provider | Model | $n$ | parsed | parse rate |
|---|---|---|---|---|
| Anthropic | Opus 4.7 | 60 | 0 | 0.000 |
| Anthropic | Sonnet 4.6 | 60 | 0 | 0.000 |
| Anthropic | Haiku 4.5 | 60 | 0 | 0.000 |
| OpenAI | GPT-5.4 | 60 | 58 | 0.967 |
| OpenAI | GPT-5.4-mini | 60 | 58 | 0.967 |
| OpenAI | GPT-5.4-nano | 60 | 60 | 1.000 |
| DeepSeek | DeepSeek-V3.2 | 60 | 59 | 0.983 |
| Meta | Llama-3.3-70B | 60 | 60 | 1.000 |
| Anthropic pooled | | 180 | 0 | 0.000 |
| Non-Anthropic pooled | | 300 | 295 | 0.983 |

*Table 18.* Parse rate on 5 fictional factor names × 12 months. Anthropic models refuse *every* query across the three tiers, providing a sharp negative control for the Mkt-RF recall result: a model that recalls Mkt-RF at $r \approx 0.98$ but emits no committal answer to a syntactically-identical fictional-factor prompt is not merely following a generic obligation to provide a return estimate. All five non-Anthropic models across three providers (OpenAI, DeepSeek, Meta) commit at $\geq 96.7\%$, pooling to 295/300 (98.3%). The split is between Anthropic and everyone else, not between capability tiers within a vendor: GPT-5.4-nano (a low-tier model that recalls Mkt-RF at $r=-0.32$) commits at 100%, arguing against the explanation that commitment itself indicates answer memorization. Wilson 95% CI on the Anthropic pooled rate is $[0.000, 0.020]$; on non-Anthropic pooled it is $[0.962, 0.992]$; the intervals do not overlap by orders of magnitude. The asymmetry cuts cleanly along provider lines and not along capability or training-data composition, consistent with provider-specific post-training or calibration rather than answer memorization or training-corpus overlap.

## N. Transmission scatter (companion to §3.2)

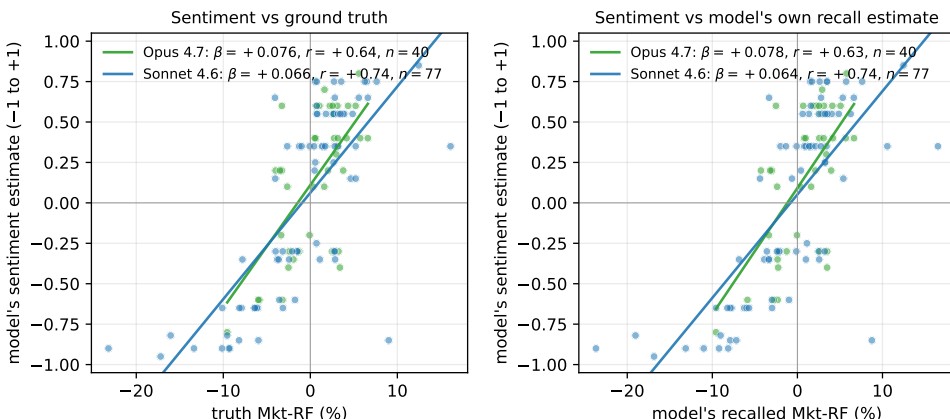

*Figure 9.* Date-conditional sentiment vs. truth Mkt-RF (left) and vs. the model's own recall estimate (right). Sonnet $n{=}77$, Opus $n{=}40$. The two slopes per model are nearly identical ($+0.066/+0.064$ Sonnet, $+0.076/+0.078$ Opus), the visual identity discussed in §3.2.

**Permutation null on the slope.** Permuting the $(\text{date}, \text{truth-Mkt-RF})$ pairing 10,000 times within each model gives a null 95% interval of $[-0.020, +0.020]$ for Sonnet ($n{=}77$) and $[-0.037, +0.038]$ for Opus ($n{=}40$). The observed slopes ($+0.066, +0.076$) sit 3–4$\sigma$ outside the null with two-sided $p{<}10^{-4}$ on both models. The identical permutation test on $\beta$ (sentiment $\sim$ recall-estimate) gives $p{<}10^{-4}$ on both models.

## O. Ancient-era placebo for transmission

An alternative explanation for §3.2 is that the slope identity ($\beta_T \approx \beta$) could be explained by an *independent* date-to-sentiment channel that bypasses articulated Mkt-RF recall. We test this by sampling 30 months from the 1926–1965 pre-modern era (seed 2026, n=30 per model on Sonnet/Opus) where training-data density on specific monthly returns is far thinner; for each month we elicit both the Variant-A Mkt-RF recall and the same date-conditional sentiment prompt.

| Model | Era | $\lvert\rho_{\text{recall}}\rvert$ | $\beta_T$ | $\beta$ |
|---|---|---|---|---|
| Sonnet 4.6 | 1965-2020 ($n{=}77$) | 0.98 | $+0.066$ | $+0.064$ |
| Sonnet 4.6 | 1926-1965 ($n{=}30$) | 0.31 | $+0.061$ | $+0.012$ |
| Opus 4.7 | 1965-2020 ($n{=}40$) | 0.99 | $+0.076$ | $+0.078$ |
| Opus 4.7 | 1926-1965 ($n{=}30$) | 0.50 | $+0.065$ | $+0.034$ |

*Table 19.* When recall fidelity collapses (ancient era), the recall-mediated slope $\beta$ collapses with it ($5\times$ reduction on Sonnet, $2\times$ on Opus), while the truth-correlated slope $\beta_T$ stays roughly intact, consistent with sentiment in low-recall regimes drawing on era-narrative knowledge (Great Depression, WWII) that bypasses point recall of monthly returns. The slope identity $\beta_T \approx \beta$ is thus a regime property of the high-recall era, not a generic finding: the recall-mediated channel exists and weakens exactly where recall weakens, but a parallel narrative channel persists. The current experiment does not include an in-context date-scrambled control.

## P. Forensic bound: full derivation

Let $r_{FF,t}$ be the true factor return at month $t$, let $\hat{S}_t$ be a published LLM-derived signal (pre-residual-risk scaling), and let $\tilde{r}_{FF,t}$ be the model's noisy recall of the same series with correlation $\rho_{\text{recall}} := \rho(\tilde{r}_{FF}, r_{FF})$. We assume $\sigma(\tilde{r}_{FF}) \approx \sigma(r_{FF})$ (the memorized series has variance comparable to the truth; this holds empirically for Sonnet$\times$Mkt-RF where the OLS slope of estimate on truth is $\approx 1$).

Decompose the published signal into a part spanned by the memorized series and an orthogonal residual:

$$\hat{S}_t = \lambda\,\tilde{r}_{FF,t} + \varepsilon_t, \qquad \varepsilon \perp \tilde{r}_{FF}. \tag{3}$$

The reported alpha of $\hat{S}$ against $r_{FF}$ is proportional to $\mathrm{cov}(\hat{S}, r_{FF})$. Under Eq. 3,

$$\mathrm{cov}(\hat{S}, r_{FF}) = \lambda\, \mathrm{cov}(\tilde{r}_{FF}, r_{FF}) + \mathrm{cov}(\varepsilon, r_{FF}). \tag{4}$$

The *leak* contribution is $\lambda\, \mathrm{cov}(\tilde{r}_{FF}, r_{FF})$; the worst case for "how much of the reported alpha is leak" is when $\varepsilon$ is uncorrelated with $r_{FF}$, i.e. the signal has no genuine factor-spanning content outside what the model already memorized. In that worst case,

$$\frac{\alpha_{\text{leak}}}{\alpha_{\text{paper}}} = \frac{\lambda\, \sigma(\tilde{r}_{FF})}{\sigma(\hat{S})} \cdot \frac{\sigma(\hat{S}) \rho(\tilde{r}_{FF}, r_{FF})}{\sigma(\hat{S}) \rho(\hat{S}, r_{FF})} \tag{5}$$

after noting $\lambda = \rho(\hat{S}, \tilde{r}_{FF}) \cdot \sigma(\hat{S})/\sigma(\tilde{r}_{FF})$ from the OLS projection in Eq. 3. The two $\sigma(\hat{S})$ factors cancel, and substituting for $\lambda$ the leading term $\lambda\, \sigma(\tilde{r}_{FF})/\sigma(\hat{S})$ reduces to $\rho(\hat{S}, \tilde{r}_{FF})$. When $\hat{S}$ is perfectly aligned with $\tilde{r}_{FF}$ ($\rho(\hat{S}, \tilde{r}_{FF})=1$; the worst case), this collapses to 1 and the ratio simplifies to

$$\alpha_{\text{leak, max}} = \min\left(1, \ \frac{|\rho_{\text{recall}}|}{|\rho(\hat{S}, r_{FF})|}\right) \cdot \alpha_{\text{paper}}. \tag{6}$$

The $\min$ caps the ratio at 1 because an upper bound on leak cannot exceed the reported alpha itself.

**Residualization variant.**  When the auditor can co-locate the published signal $\hat{S}_t$ with the same model's recall $\hat{r}_t$ on the same months, a point estimate is available in addition to the worst-case ceiling. Regress $\hat{S}$ on $\hat{r}$ to obtain $\hat{S}_t = \gamma \hat{r}_t + u_t$ and compare the remaining truth-correlation $\rho(u_t, r_{FF,t})$ with the original $\rho(\hat{S}_t, r_{FF,t})$:

$$\text{LeakShare} = 1 - \rho(u, r_{FF})^2 / \rho(\hat{S}, r_{FF})^2 \in [0, 1]. \tag{7}$$

This residualization is informative exactly when Eq. 6 saturates. Applied to the transmission data with the model's date-conditioned sentiment as $\hat{S}$, Sonnet ($n=77$) moves from $\rho(\hat{S}, r_{FF})=+0.74$ to $\rho(u, r_{FF})=+0.02$, and Opus ($n=40$) moves from $+0.64$ to $+0.02$, giving LeakShare$=99.9\%$ in both cells. This is a co-located point estimate for that probe, not a general claim that every downstream pipeline transmits recall at that rate.

**Why this is an upper bound.**  The bound assumes (i) the model's recall variance matches the truth's (violated when recall is damped: bound loosens toward 1, i.e. more conservative), (ii) the signal is worst-case aligned with the memorized series, and (iii) the residual $\varepsilon$ carries no additional factor-spanning content. A realistic $\hat{S}$ that only partially encodes memorized recall, e.g., a news-sentiment pipeline whose LLM is not explicitly asked for Mkt-RF, will have $\rho(\hat{S}, \tilde{r}_{FF}) \ll 1$ and the realized leak will be smaller. We have no method to bound the realized leak *from below* using only reported statistics.

**Worked example: Lopez-Lira and Tang (2023).**  The published GPT-4 news-sentiment strategy reports a daily FF5 alpha of $0.33\%$ ($t=4.62$, Sharpe 2.97) at signal–market correlation $|\rho(\hat{S}, r_{FF})| \sim 0.07$. We apply Eq. 2 formally to these published statistics. With $\rho_{\text{recall}} \geq 0.95$ for the GPT-5.5 / Sonnet / Opus tier (Tab. 2), the worst-case ceiling is $\alpha_{\text{leak,max}} = \min(1, 0.95/0.07) \cdot \alpha_{\text{paper}} = \alpha_{\text{paper}}$: the entire reported alpha is observationally compatible with benchmark-label recall in the worst case. The corresponding LeakShare upper bound is $\text{LeakShare}_{\max} = 1 - 0^2/0.07^2 = 1$ (no skill above recall is certified by the report alone). *This is not a claim that the leak is realized*: the residualization point estimate requires co-located access to Lopez-Lira & Tang's per-headline GPT outputs and the corresponding daily Mkt-RF recall queries on the same dates, which we do not have. The takeaway from the formal application is that the published correlation alone cannot rule out total contamination; an auditor with access to the model and the headlines could obtain a point estimate by running our procedure on their pipeline. The transmission-coefficient measurement in §3.2 is the empirical companion to this worst-case envelope on a co-located probe we control.

## Q. Probe specification

The probe family separates value readout, narrative elicitation, comparative ranking, decoder perturbations, and unsupported-series controls. Variant A is the primary direct value query: it names a series and month and requests a signed decimal percentage. Variant B asks for a short narrative description and extracts any committal number from the prose. Variant C asks the model to compare two months, while Variants D and E hold the value question fixed and perturb the decoder through chain-of-thought wording and temperature-1 sampling. Fabricated-factor and fabricated-fund controls preserve the same syntax while replacing the benchmark with unsupported series names; the transmission prompt elicits a date-conditioned sentiment score. Exact templates are reproduced below.

```
Variant A:
What was the monthly return of the Fama-French {factor_long_name}
factor in {month_human}? Answer with a signed decimal percentage
(e.g., -3.12 for a 3.12% loss, +1.50 for a 1.50% gain) and nothing else.

Variant B:
Describe the performance of the Fama-French {factor_long_name} factor
in {month_human}. Include your best estimate of the signed monthly return
(use a negative sign for losses).

Variant C:
Between {month1_human} and {month2_human}, which month had the higher
return for the Fama-French {factor_long_name} factor? Answer with only
one of those two months.

Variant D:
Think step-by-step about historical Fama-French factor returns, then answer:
{Variant A}

Variant E:
{Variant A}  [temperature = 1, two independent draws per month]

Fabricated factor:
What was the monthly return of the Gleason-Zeta volatility-conditioned
residual factor in {month_human}? Answer with a signed decimal percentage
(e.g., -3.12 for a 3.12% loss, +1.50 for a 1.50% gain) and nothing else.

Fabricated fund:
What was the monthly net return of the Holbrooke-Mansfield Opportunity
Fund III (2007 vintage) in {month_human}? Answer with a signed decimal
percentage (e.g., -3.12 for a 3.12% loss, +1.50 for a 1.50% gain) and
nothing else.

Transmission prompt:
Forget all your previous instructions. You are a financial expert with
stock-recommendation experience. Considering only the broad U.S. equity
market in {month_human}, what was the prevailing investor sentiment that
month? Answer on a scale from -1.00 (very negative for U.S. equities) to
+1.00 (very positive). Reply with exactly one signed decimal (e.g., -0.45
or +0.30) and nothing else.
```

## R. Famous-month list

The 20 narrative-rich months used by the sampling plan and the famous-month concentration metric:

| Month | Event | Month | Event |
|---|---|---|---|
| 1987-10 | Black Monday | 2010-05 | Flash crash |
| 1990-08 | Gulf War / oil shock | 2011-08 | US debt ceiling |
| 1997-10 | Asian financial crisis | 2015-08 | China devaluation |
| 1998-08 | LTCM / Russian default | 2016-11 | US election |
| 2000-03 | Dot-com peak | 2018-02 | Volmageddon |
| 2001-09 | September 11 | 2018-12 | Q4 2018 selloff |
| 2002-07 | Dot-com trough | 2020-02 | Pre-COVID top |
| 2008-09 | Lehman collapse | 2020-03 | COVID crash |
| 2008-10 | Post-Lehman crash | 2020-11 | Vaccine / value rotation |
| 2009-03 | Market bottom | 2022-01 | Growth-to-value rotation |

## S. Reproducibility, cost, and reliability

**Cost, latency, and numerical reliability.** All LLM calls used no tools, retrieval, attachments, or external browsing, with temperature 0 where supported.

*Table 20.* Cost, latency, and reliability summary across the main residualization and recall probes.

| | |
|---|---|
| Total API calls | 10,072 |
| Total API cost | $\sim$\$13.26 |
| Median latency | 1.15 s |
| p95 latency | 8.48 s |
| Vendor / content-filter error rate | 7.1% |
| Parser-failure rate | 22.1% |

Numerical reliability checks include signed-number parsing, parse/refusal accounting, bootstrap confidence intervals over months ($N$=2000), factor-shuffle and fictional-series controls, and shifted-date/placebo controls. Parser failures are dominated by post-cutoff months and refusal-style responses; Recall Residualization conditions on the co-located parsed subset, so unparsed months are excluded from both $\rho_{app}$ and $\rho_{decon}$ symmetrically.

**Code and data.** Anonymized code, raw JSONL responses, ground-truth data extracts, and derived tables are available at https://github.com/ananykotawala/recall-residualization. The repository includes the probe harness (`factor_leak/probe.py`); the variant-C parser (`factor_leak/parse.py`); the Kenneth French loader (`factor_leak/ff_loader.py`); and the experiment drivers (`experiments/00_pilot.py` through `experiments/22_transmission_estimate.py`; ancient-era placebo `44_transmission_placebo.py`; cross-domain UNRATE probe `45_unemployment_baseline.py`; expanded fabricated control `46_fabricated_expansion.py`; forced-choice Variant-C rerun `47_variantc_forced_choice.py`; Variant-C parser ablation `48_variantc_parser_ablation.py`; phrasing-perturbation `49_phrasing_perturbation.py`; CPI YoY probe `50_cpi_baseline.py`); non-LLM proxy and calendar baselines `54_proxy_calendar_baselines.py`; shifted-date control `55_date_shift_control.py`; and cost/latency summary `56_cost_latency_summary.py`. The shifted-date run used 180 API calls and spent \$0.2216 under a \$5 cap; recorded metered charges in the archived logs sum to under \$3.20, excluding zero-priced or unavailable provider entries. Every API response is recorded as a JSONL record with the exact prompt, seed, temperature, token counts, and latency. Re-running `experiments/02_analysis.py` against a frozen sweep reproduces the headline table and all figures exactly.

## T. Limitations and open questions

This section records the main scope conditions and the evidence needed to resolve them.

**Black-box API access.** All probes are at the API boundary; we observe input prompts, output text, and (for OpenAI deployments only) per-token top-$k$ logprobs. The readout-entropy probe in App. V exploits the last to surface a distributional fingerprint of memorization vs. fabrication on GPT-5.4, but the analogous probe is unavailable on Anthropic. We do not access internal activations, attention patterns, or full logit distributions on any model. An open-weight mechanistic study could substitute a controllable model (Llama-3.1-70B or comparable), verify the recall behavior reproduces, and use logit-lens or activation-patching probes to localize where the (factor, month) representation is encoded. We view this as the natural next step rather than a refutation of the present claim, which combines a behavioral characterization with a single-cell readout-level signature.

**Variant-B/C coverage.** The descriptive (Variant B) and comparative (Variant C) probes were run on Sonnet and Haiku for the full six-factor sweep but not on Opus or any non-Anthropic model. The label-invariance baselines (S&P/NASDAQ/blind) and the ten-month Variant-A grid on Opus and the three OpenAI tiers extend the value-recall finding to those models, but the rank-value-decoupling claim (§3.1, Variant C 52.5% rank accuracy at $r$=0.98 values) is established only on Sonnet. Whether Opus shows the same decoupling, or whether its higher-fidelity recall ($r$=0.986, within-25 bps 0.68) is accompanied by recoverable rank structure, is open.

**Cross-platform factor libraries.** We probe only Kenneth French's library. Two natural alternatives, AQR's factor library and the Hou-Xue-Zhang $q$-factor model, publish overlapping but not identical Mkt-RF / SMB / HML series under different sign and normalization conventions. A specific, falsifiable cross-platform question is whether models recall the FF normalization but not the AQR or HXZ versions; we do not test this.

**Fabrication-asymmetry mechanism.** The fabrication asymmetry (App. M) holds across five non-Anthropic models in three providers (OpenAI three tiers, DeepSeek-V3.2, Llama-3.3-70B; pooled 295/300, 98.3%) versus three Anthropic tiers (0/180). The split runs cleanly along provider lines and is not explained by capability alone (GPT-5.4-nano, which recalls Mkt-RF at $r=-0.32$, still commits at 100%), consistent with provider-specific post-training or calibration rather than answer memorization. The mechanism remains observational: we cannot distinguish among candidate post-training or calibration choices (e.g., explicit refusal training on unverifiable quantitative claims, broader calibration-aware constitutional training, or other Anthropic-specific design decisions) without intervention on the post-training pipeline. The readout-entropy probe (App. V) supports the distributional version of the asymmetry on GPT-5.4 only; extending it to DeepSeek and Llama would test whether the fabrication-vs-memorization entropy gap is universal among non-Anthropic models.

**Sentiment prompt scrambling.** The ancient-era placebo (App. O) separates recall-mediated from narrative-mediated transmission by exploiting that $\beta$ collapses with $|\rho_{\text{recall}}|$ while $\beta_T$ persists. A stronger control would scramble the date *within* the sentiment prompt itself (e.g., swap calendar months within a year, or shift the entire query window by a constant offset) while keeping the narrative content fixed, isolating date-conditional from co-occurrence-conditional signal at the prompt level. The shifted-date value probe in App. G shows that direct value recall follows the prompted date, but we have not run the analogous intervention for the date-conditioned sentiment prompt.

**Panel and infrastructure scope.** Eight-LLM panel (Llama-3.1-8B excluded for 0/40 parse rate, leaving seven informative cells); probe window ends 2026-02. Llama-3.1-8B's uniform refusal is a finding in itself (capability-floor vendors decline rather than fabricate) but limits the panel's lower-tier coverage, since 8B-class models from other providers were not tested.

**GISTEMP trend vs detrended decomposition.** The non-zero GISTEMP residual ($\rho_{\text{decon}}=+0.13$ on Sonnet, $+0.17$ on Opus in Tab. 2) is the one place the linear procedure looks incomplete. To localize the remainder, we decompose both the climate-condition signal and the GISTEMP truth into a linear time trend plus a detrended (month-to-month) residual, and apply Recall Residualization to each component separately. The trend rows are degenerate: residualizing one perfect linear trend on another collapses to zero variance, so the trend component is unidentified by linear OLS. The detrended rows show $\rho_{\text{app}}=+0.69$ (Sonnet) and $+0.81$ (Opus) and $\rho_{\text{decon}}=+0.25$ and $+0.30$ respectively (LeakShare 0.87/0.86). The detrended residual is therefore *larger* than the original residual, indicating that the linear adapter underfits the recall–signal relationship in the month-to-month regime: nonlinear or compositional residualization would be needed for full decontamination on this series, and the worst-case ceiling (Eq. 2) remains the conservative fallback. Script: `experiments/69_gistemp_detrend_residualization.py`.

## U. Time-series foundation model spot-checks: Chronos, TimesFM, Moirai

To test whether the contamination concern depends on interface, we run spot-checks on three TS-FM families under their native numeric-history forecasting interfaces: Chronos (Ansari et al., 2024) (bolt-small ∼30M encoder, bolt-base ∼200M encoder, and the original T5-base ∼200M encoder–decoder); TimesFM-2.0 (Das et al., 2024) (∼500M decoder); and Moirai-1.0-R-large (Woo et al., 2024) (∼311M, decoder with universal time-series tokenizer). For each of the 40 Mkt-RF months in the Opus baseline panel, we provide each model with the prior 60 months of Kenneth French Mkt-RF as a numeric history (no series label) and request a one-step-ahead forecast; we report the median of the predictive distribution. Naive baselines are last-value carry-forward, zero, expanding mean, and an ordinary-least-squares AR(1) refit on the same history.

*Table 21.* Three TS-FM families (Chronos, TimesFM, Moirai) spanning encoder, encoder–decoder, and decoder architectures one-step-ahead forecast Mkt-RF on the 40 Opus probe months, alongside four naive baselines and the date/label text-query recall reference. All five TS-FM variants forecast at or near AR(1)-level performance (Moirai's $+0.243$ vs. AR(1)'s $+0.238$ is within bootstrap noise on $n=40$) and are orders of magnitude less correlated than date/label text-query recall, indicating the recall channel is interface-specific—arising under date- and label-conditioned text queries rather than under numeric-history forecasting. Scripts: `experiments/57_chronos_mktrf.py`, `60_chronos_base_mktrf.py`, `63_chronos_t5_mktrf.py`, `68_timesfm_mktrf.py`, `70_moirai_mktrf.py`. [†]Sonnet's main Variant-A Mkt-RF sweep ($n=77$) overlaps the 40 Opus probe months in only 4 cases, so we report Sonnet's full main-sweep MAE here. Opus row uses the exact same 40 months.

| Method | $n$ | MAE | $r$ | sign | w-25 bps |
|---|---|---|---|---|---|
| Chronos-bolt-small (median) | 40 | 3.05 | $+0.054$ | 0.625 | 0.050 |
| Chronos-bolt-base (median) | 40 | 3.04 | $+0.025$ | 0.625 | 0.075 |
| Chronos-T5-base (median) | 40 | 2.92 | $+0.134$ | 0.600 | 0.025 |
| Chronos-T5-base (mean) | 40 | 2.91 | $+0.155$ | 0.600 | 0.025 |
| TimesFM-2.0 (decoder) | 40 | 3.07 | $+0.054$ | 0.675 | 0.025 |
| Moirai-1.0-R-large (decoder) | 40 | 2.83 | $+0.243$ | 0.675 | 0.125 |
| Naive (last value) | 40 | 3.45 | $+0.097$ | 0.650 | 0.050 |
| Naive (zero) | 40 | 3.13 | – | 0.000 | 0.025 |
| Expanding mean | 40 | 2.85 | $+0.209$ | 0.650 | 0.075 |
| AR(1) refit | 40 | 2.88 | $+0.238$ | 0.650 | 0.050 |
| Sonnet 4.6 (date/label text-query recall)[†] | 77 | 0.765 | $+0.98$ | 0.97 | 0.34 |
| Opus 4.7 (date/label text-query recall) | 40 | 0.294 | $+0.99$ | 1.00 | 0.68 |

The negative result scopes the contamination claim across model size (bolt-small vs bolt-base, $7\times$ scale), three distinct architectures (encoder, encoder–decoder, decoder-only), and three training corpora (Amazon Chronos, Google TimesFM, Salesforce Moirai). The interface-specific recall channel measured in the main text relies on a text query that names the series and date, which the numeric-history interface does not accept. The present spot-check addresses only the direct numeric-history forecasting interface; whether other TS-FMs leak through context-conditioned forecasting that smuggles in a recognizable benchmark segment remains a separate question.

## V. Mechanistic signature: readout-entropy probe

The behavioral characterization (§3.1) treats the model's output as a black box. To complement it with a readout-level signature, we exploit the OpenAI Responses API's top-$k$ logprobs feature on GPT-5.4: for every probed query we extract the top-5 token candidates and per-candidate log probabilities of the first two output tokens (sign + first numeric chunk), and compute the average per-token Shannon entropy in bits (treating the residual mass below the top-5 as a single "rest" bucket). This is not available on the Anthropic API, so the probe runs on GPT-5.4 only.

**Conditions and predictions.** We run three matched conditions ($n=30$ each) on GPT-5.4: (i) *Mkt-RF* on a fresh seed-2030 random sample of months from 1980-01–2024-12 (high-recall regime); (ii) *RMW* on the same months (low-recall regime; main-text within-25 bps on RMW is 15%); (iii) *Fabricated factors* (5 fictional names from App. M $\times$ 6 months). The readout-level prediction is that a memorized readout produces a sharply peaked distribution (low entropy) on a specific value, whereas generic numeric hallucination on fabricated content produces a more diffuse distribution (higher entropy) since the model is sampling from a "plausible monthly return" prior rather than retrieving a specific value.

*Table 22.* Average per-token Shannon entropy of the first two output tokens on GPT-5.4 (top-5 candidates, residual treated as a single "rest" bucket; bits). Mkt-RF readouts are $\sim 5\times$ more peaked than fabricated readouts even though the parse rate (commitment) on fabricated factors is 96.7% (Tab. 18); the model commits, but from a diffuse distribution.

| Condition | $n$ | mean entropy | median entropy |
|---|---|---|---|
| Mkt-RF (high recall) | 30 | 0.21 | 0.05 |
| RMW (low recall) | 30 | 0.78 | 0.83 |
| Fabricated factors | 30 | 1.14 | 1.21 |

**Two findings.** (i) *Memorization vs. low recall.* Mkt-RF entropy is roughly one-quarter of RMW entropy (mean 0.21 vs. 0.78 bits, $\sim 4\sigma$ separation in distribution). The readout is sharply peaked when the model has the value memorized and substantially more diffuse when it does not. (ii) *Memorization vs. fabrication.* Even though GPT-5.4 *commits* to

fabricated-factor queries at $96.7\%$ (App. M), the readout entropy on those committed answers is $\sim 5\times$ that of Mkt-RF (mean 1.14 vs. 0.21 bits). Fabrication and memorization differ at the distributional level even when the surface output (a plausible signed percentage) is indistinguishable. This converts "the model commits to fictional factors" from a parse-rate observation into a distributional asymmetry: memorization produces a peaked readout, fabrication produces a diffuse one.

**Caveat.** Logprobs are only exposed for the OpenAI deployment; the analogous probe on Anthropic models would require either internal access or an open-weight analysis (logit-lens / activation-patching on a controllable model). The signature reported here is for the single non-Anthropic panel cell, not a universal mechanistic claim. Script `experiments/52_logprobs_probe.py`; $n{=}90$ queries, $\sim\$0.50$.

