# OpenReview forum: "Recall Residualisation: Decontaminating Foundation-Model Evaluation on Public Time-Series Benchmarks"
_ICML.cc/2026/Workshop/FMSD — FMSD @ ICML 2026 Poster_

### Official Review · Reviewer_DyJk · 2026-05-20
**Quantifying Benchmark Contamination in Date-Relevant LLM Prediction**

**Rating:** 7
**Confidence:** 3

**Review:**

# Summary

This paper analyses the apparent contamination in evaluating contextual foundation models due to memorized historic knowledge (recall) rather than true predictive ability. They showed that many LLMs (Sonnet 4.6, Opus 4.7, GPT-5.5, Llama-4, and DeepSeek-V3.2) were able to recall with high correlation a historic value given just date and problem context. The authors propose Recall Residualization that compares LLM performance when given historical context \- previous values \- in addition to the date and problem context with just recall correlation. They found that LLMs heavily rely on recalling the correct value via public data given the date rather than having a predictive ability.

# Strengths

* LeakShare effectively compares performance from decontaminated prediction via OLS with the apparent contaminated prediction.
* Paper includes extensive ablations indicating that results are robust to prompting and other effects.
* Workshop audiences would find the paper’s analysis on benchmark contamination to be of interest.

# Areas for Improvement \+ Detailed Comments

* In section 3.3 the authors evaluate existing TSFM as numeric-only counterparts. The models (Chronos-T5-base, TimesFM-2.0, Moirai-1.0) are no longer SOTA models in the field. While I expect similar results with newer versions of the used models (i.e. Chronos-2, TimesFM-2.5, Moirai-2.0), it would be a fairer comparison given how modern the LLMs in the paper are (e.g. Opus 4.7 and GPT 5.5 being from April of this year while Chronos-T5 was from March of 2024).
* As mentioned in Appendix G, the LLMs rely heavily on the dates, where shifting the prompted date but fixing the historic values resulted in strong predictions for the shifted date but not relative to the original historic values. I am curious to see how the decontamination scores compare to prompting the LLMs either without a date or claiming the date is in the future.

---

### Official Review · Reviewer_d3tx · 2026-05-21
**The paper proposes a methodoloy to measure data leakage in time series data from public sources.**

**Rating:** 6
**Confidence:** 4

**Review:**

## Summary

The algorithm uses LLM queries to ask for the value of the series at a specific point in time to reconstruct the series, then regresses the original series on the LLM-generated one to compute correlations. For TSFM, reconstruction is achieved by providing historical values and producing forecasts. The LLMs show a strong correlation, indicating, according to the paper, evidence of leakage. TSFMs show less correlation.

## Strenghts

- The proposed algorithm is general enough to allow analysis from LLMs and TSFMs.
- The discussion of the problem is well presented.
- Experiments seem to be legit.

## Areas for Improvement

- One could stress the argument and say that LLMs don't suffer from data leakage, but that they are very good forecasters of these series. How can we be sure that there is evidence of leakage in the first place?
- How about using other metrics, such as MAE, RMSE, or MASE, and not just correlation?
- Plots of regenerated series by LLMs and TSFMs can improve the overall exposition a lot.
- Although at some point the paper claims that traditional models, such as Arima, don't need to be exposed to this process, they are not included in the analysis. Its inclusion can help to enhance the results.